# Harmonizing sound and light: X-ray imaging unveils acoustic signatures of stochastic inter-regime instabilities during laser melting

Milad Hamidi Nasab [1] ✉, Giulio Masinelli [2] ✉, Charlotte de Formanoir[1], Lucas Schlenger[1], Steven Van Petegem [3] ✉, Reza Esmaeilzadeh[1], Kilian Wasmer [2], Ashish Ganvir[4], Antti Salminen [4], Florian Aymanns [5], Federica Marone [6], Vigneashwara Pandiyan [2], Sneha Goel [3] & Roland E. Logé [1]

Laser powder bed fusion (LPBF) is a metal additive manufacturing technique involving complex interplays between vapor, liquid, and solid phases. Despite LPBF's advantageous capabilities compared to conventional manufacturing methods, the underlying physical phenomena can result in inter-regime instabilities followed by transitions between conduction and keyhole melting regimes − leading to defects. We investigate these issues through operando synchrotron X-ray imaging synchronized with acoustic emission recording, during the remelting processes of LPBF-produced thin walls, monitoring regime changes occurring under constant laser processing parameters. The collected data show an increment in acoustic signal amplitude when switching from conduction to keyhole regime, which we correlate to changes in laser absorptivity. Moreover, a full correlation between X-ray imaging and the acoustic signals permits the design of a simple filtering algorithm to predict the melting regimes. As a result, conduction, stable keyhole, and unstable keyhole regimes are identified with a time resolution of 100 μs, even under rapid transitions, providing a straightforward method to accurately detect undesired processing regimes without the use of artificial intelligence.

Laser powder bed fusion additive manufacturing (LPBF AM), currently the most widely adopted metal additive manufacturing process, is a technology capable of directly producing intricate three-dimensional metallic components from digital Computer-Aided Design (CAD)

models. Specifically, a conventional LPBF process utilizes a high-power-density laser to scan 2D patterns over a flat bed of microscopic (~15–100 μm) metal powder, creating melt pools in the order of 100 μm wide in each successive layer of powder to manufacture a desired 3D

[1]Thermomechanical Metallurgy Laboratory—PX Group Chair, École polytechnique fédérale de Lausanne (EPFL), CH-2002 Neuchâtel, Switzerland. [2]Laboratory for Advanced Materials Processing (LAMP), Swiss Federal Laboratories for Materials Science and Technology (Empa), CH-3602 Thun, Switzerland. [3]Structure and Mechanics of Advanced Materials, Photon Science Division, Paul Scherrer Institut, PSI, Forschungsstrasse 111, 5232 Villigen, Switzerland. [4]Digital Manufacturing and Surface Engineering (DMS), Department of Mechanical and Materials Engineering, University of Turku, FI-20014 Turun yliopisto, Finland. [5]EPFL Center for Imaging, École polytechnique Fédérale de Lausanne (EPFL), Lausanne, Switzerland. [6]Swiss Light Source, Paul Scherrer Institute, 5232 Villigen, Switzerland. ✉e-mail: milad.hamidinasab@kuleuven.be; giulio.masinelli@epfl.ch; steven.vanpetegem@psi.ch

solid part[1]. However, despite LPBF's unparalleled capabilities—such as the ability to produce complex geometries, custom parts, and open-cell structures (3D lattices) with limited material waste in the metal-AM domain—it suffers from several limitations that endanger its widespread adoption in demanding industrial sectors (i.e., aerospace, automobile, and medical). These limitations emerge from uncommon thermal history and rapid solidification, along with defects randomly introduced during the process resulting in a lack of repeatability and robustness. Additionally, the complex interactions between the high-power-density laser source and the powder bed bring about entangled interdependencies between the considerable number of process parameters, whose effects are not yet fully comprehended. Furthermore, significant melt pool geometry variations are observed even under constant process parameters, revealing problematic in-process instabilities. These variations can take place due to several factors: (i) the effect of scanning strategy and corresponding heat build-up in intra-layer laser scan[2,3]; (ii) the acceleration and deceleration at the beginning and the end of the laser track, resulting in an increased laser dwell time and uncontrolled localized energy density[4]; (iii) the presence of geometrical features (e.g., overhangs and walls) diminishing the heat sink effect[5,6]; and (iv) the dependency of laser absorptivity on the angle of incidence and the resultant non-uniform laser absorption (Fresnel absorption) and temperature distribution alongside the scanning vector[7]. Moreover, in the event of large melt pool geometry variations, inter-melting regime instabilities might occur, with occasional or regular transitions between conduction and keyhole regimes[6].

In recent years, the mentioned process instabilities started to be investigated thanks to the emergence of high-speed synchrotron X-ray imaging, bringing specific attention to the keyhole melting mode, recognized as one of the main responsible phenomena in the formation of a ubiquitous porosity[7–9]. This regime is the product of excessive energy density beyond a criterion[10,11], bringing about rapid evaporation of the molten metal in the laser-material interaction zone and the formation of recoil pressure. The intense recoil pressure pushes the surrounding liquid metal downwards by overcoming the surface tension of the molten metal, generating a profound and high aspect-ratio cavity called a keyhole[12]. Upon the formation of this narrow cavity, the keyhole becomes a gray body due to a dramatic increase in laser absorption via multiple reflections on its walls, further contributing to its energy absorption efficiency[13,14]. Furthermore, intertangled complex thermo-mechanical interactions such as thermocapillary force, Marangoni convection, recoil pressure, and gas plasma call forth keyhole instabilities[15–17], subjecting keyhole to axial fluctuations and radial perturbations[7,11,12,18], significantly increasing the chance of keyhole collapse and the entrapment of gas bubbles in the melt[4,8,19]. Upon solidification, these bubbles might get pinned and form porosity in the final consolidated part, acting as stress raisers, becoming detrimental to fatigue[20] and other mechanical properties of parts[21,22].

Consequently, the stochastic inter-regime instabilities can result in the formation of undesired defects and significant alterations in local thermal history and melt pool geometry, partaking into the microstructural inhomogeneity of the part. Nonetheless, LPBF must yield products with predictable mechanical properties to ensure its ubiquitous implementation in strategic industries. Therefore, the widespread application of real-time/in situ quality monitoring techniques becomes paramount to improving the reliability of the LPBF process. In the past 10 years, various sensors (e.g., optical (photodiodes and pyrometers), vision (high-speed and thermal imaging), and acoustic emission (AE)) have been implemented into the LPBF process for this aim[23,24]. The principal advantage of in situ monitoring techniques over post-mortem approaches is that upon detection of undesirable events, the process can be stopped, and appropriate healing measures via re-melting of the top layer can be taken, removing the defects and saving resources[25,26]. However, most of the in-situ

monitoring techniques provide information limited to the top surface region of the process zone, coming short on the complex phenomena taking place at the bottom of the depression zone[27]. Complementary to these approaches, AE, as a low-cost and robust monitoring technique, might be up to the task—as it can provide information relevant to melt-pool dynamics and its volumetric behavior[27–29]. During the LPBF process, a significant amount of noise is generated, which can be utilized as an in situ feedback signal[29–31]. In the context of laser welding, acoustic waves can originate from three primary sources: (1) acoustic waves induced within the substrate through phase transformation phenomena, including solid-state, solid/liquid, and liquid/gas transformations[32,33], (2) thermal stress-induced acoustic waves in the laser-material interaction zone; and (3) acoustic waves resulting from variations in gas momentum due to heating, boiling processes, and subsequent changes in gas pressure[33,34]. The latter can occur due to various phenomena in the laser-material interaction zone, such as metal vaporization, formation and expansion of a microjet at the bottom of the keyhole, pressure variation caused by surface tension gradient around the keyhole opening due to thermal deformation[34], the collision of liquid with the keyhole rear wall due to a drop in vapor pressure in the lower part of the cavity, collapse of liquid against the melt pool boundaries, penetration of the microjet into the melt and its impact on the surface of formed keyhole porosities, pore collapse, and subsequent rebound, and periodic over-heating induced explosion at the bottom of the keyhole after its collapse[8,35–38]. Collectively, these events can contribute to the periodic oscillation of the keyhole within a reported frequency range of 20–50 kHz[39].

Attempted works on the use of AE in the LPBF process mainly concentrated on the classification of process regimes[27,40] or specimen quality[41–43] by applying machine learning (ML) algorithms to acoustic signals processed by Fourier or wavelet transforms. These works investigated the effect of different sets of process parameters on the melting regime, the corresponding acoustic signal, and the quality of the final part in terms of porosity. A limitation, however, is that artificial intelligence (AI) approaches can result in a model that learns to detect changes in the process parameters rather than the process regime and defect formation[35,43]. For instance, a recent work focused on the detection of keyhole porosities via synchronization of the acoustic signal with the position of the laser at each instance in time, enabling the ground truth labeling of defect locations[44]. An ML algorithm was, in this case, employed to detect the porosity formation based on statistical features extracted from the partitioned acoustic data. However, there was no thorough investigation of the melting regime and no inquiry into whether the algorithm detects the ever-existing effect of process parameters or the actual formation of defects. According to these reports, acoustic emissions in the range of 10–40 kHz carry significant information on the keyhole regime[40,44], in-line with the oscillation frequencies reported from operando X-ray imaging of the LPBF process[39]. Identifying single events of defect formation or regime changes under constant process parameters can shed light on the acoustic signatures of these events and lead the way to develop a more robust AE-based monitoring approach—an approach that has not yet been addressed in the literature. A combination of operando high-speed X-ray imaging and implementation of synchronized in situ acoustic measurements could reveal the critical relationship between the physical events and the corresponding acoustic signal frequency contents, shedding light on previously unexplored insights.

In this work, we aim to delve deeper into the LPBF process by focusing on the identification of individual events of defect formation and regime changes, particularly under constant process parameters. This approach seeks to uncover specific acoustic signatures associated with these events, enhancing the robustness of AE-based monitoring—an aspect that has received limited attention in the existing literature. In our methodology, we introduce the simultaneous synchronized use of operando high-speed X-ray imaging and in-situ acoustic

measurements. Specifically, we observe inter-regime instabilities in LPBF fabricated structures under constant process parameters and conditions using operando high-speed transmission X-ray imaging. Concurrently, integrating an optical microphone into the LPBF building chamber allows us to capture in-situ airborne acoustic emission signals during the process. This strategy enables us to reveal the critical relationship between physical events and their corresponding acoustic signal frequency contents. Distinct from traditional machine learning classification models[27], our work adopts a segmentation-based approach that enables continuous, probabilistic assessment of the melting regime throughout the process, not just a discrete classification. Our contribution lies in the introduction of a data-driven filtering pipeline that effectively segments the acoustic emission signals into corresponding regimes, utilizing the X-ray imaging data as ground truth. This technique, although data-driven, differs fundamentally from typical machine learning or artificial intelligence algorithms. While the latter often relies on extensive training with a wide range of data, and its performance can be hindered by changes in the data distribution or highly correlated features, our method is designed to be robust under constant process parameters, with an emphasis on identifying the correlation between individual physical events and their acoustic signatures. The data-driven filtering pipeline is tailored to cope with the complex and high-dimensional LPBF process by leveraging the X-ray imaging data, which provides an explicit physical interpretation to guide the segmentation. Therefore, we believe this approach is more immune to the challenges many machine learning methods face when differentiating between highly correlated signatures from complex physical systems such as LPBF. Moreover, an essential advantage of our approach is the simplicity of the filtering pipeline, which enables a physically-backed interpretation of the results. This straightforwardness paves the way for the development of robust acoustic emission-based in-situ monitoring systems capable of accurately capturing stochastic regime instabilities and porosity formation.

## Results

### Operando observation of the thermo-mechanical instabilities

Operando X-ray imaging was utilized to shed light on the short-term regime instabilities that are specific to the laser melting process through the implementation of the mini-LPBF apparatus capable of mimicking it, as described in the "Methods" section. Thin walls were fabricated via the mini-LPBF device in keyhole printing conditions with a Gaussian beam at the focal point, focused on top of the powder bed with a measured beam diameter of 27.5 $\mu$m at $1/e^2$. A nominal power of 180 W, a scanning speed of 280 mm/s with a hatch distance (5 unidirectional line scans) and layer thickness of 40 and 30 $\mu$m were selected respectively (c.f. Supplementary Table 1). The laser was operated in a pulsed mode with a repetition rate of 250 kHz and a nominal temporal pulse length of 2 $\mu$s. The printing result is a wall with a height range of 640–750 $\mu$m, a thickness of about 250 $\mu$m and a length of 5 mm in the as-built condition (see Fig. 1). The fabrication of the walls in the keyhole regime brings about a large cloud of keyhole porosities distributed randomly in the bulk. Furthermore, scanning lines closer to the side edges of the wall induces the formation of larger

melt pools due to excessive feed of metal particles as a consequence of capillary forces[45], and hence, in higher heights at the longitudinal borders of the wall (see Fig. 1). A number of these metal particles remain only semi-attached to the top and side surfaces of the wall generating further stochastic geometrical inaccuracies to the wall geometry[46]. For the experiments, laser re-scanning of the top surface of the wall was done after top flat grinding of the wall by the same laser defocused to a factor of two or more in diameter, with increased nominal power and substantially reduced laser scanning speed (c.f. Supplementary Table 1). Several laser re-scanning operations were performed on each wall, modifying the wall topography, and resulting in different height values for each consecutive track. A non-uniform heat dissipation alongside the wall is generated as a result of keyhole porosities and geometrical unwanted features. Furthermore, the height variations along the wall introduce changes in the laser focus and the projected intensity profile on the top surface of the wall. These height variations, along with the residual waviness left by previous laser remelting, also contribute to variations in the laser incident angle and absorptivity along the track. Besides, utilization of pulsed mode laser could yield an increase in absorption probability, favoring contingency of regime transitions in laser remelting passes[47].

We effectively provoked the regime instability phenomenon under constant sets of process parameters, selecting a combination of parameters resulting in near-transition conduction conditions utilizing the state-of-the-art scaling laws[11]. Apart from the selection of the process parameters, we employed several diverging and converging laser beams on the (to be melted) wall surface, further away or closer to the focusing lens, respectively. Due to a small misalignment of the laser beam through the optics, a deviation of the Gaussian beam profile occurred (c.f. Supplementary Fig. 1), strongly visible in the divergent part of the beam. Nevertheless, we could always adjust the nominal laser power and scanning speed to obtain transitions between interesting melting regimes. We observed the occurrence of regime instabilities from conduction to stable and from stable to unstable keyhole under constant laser process parameters in single- and double-line scans. The instabilities could last from hundreds of microseconds up to tens of milliseconds. The operando X-ray imaging indicated that the regime instabilities might be a common phenomenon in individual scan lines, in regions initially believed to be steady state[4], i.e., far from the beginning and the end of the scanning vector.

The stable keyhole and unstable keyhole were distinguished based on observable characteristics in the X-ray images. A stable keyhole is characterized by a continuous, well-defined shape and the absence of significant fluctuations or instabilities. On the other hand, an unstable keyhole exhibits periodic collapse and the formation of porosities. These instabilities are visible as changes in the keyhole shape, such as irregular boundaries, fluctuating depths, and the presence of porosity formations. To determine the transition from stable to unstable keyhole, we focused on the first appearance of porosity resulting from a keyhole collapse. The appearance of porosities indicates the onset of instability in the keyhole, marking the transition to an unstable regime. Conversely, the stable keyhole regime is identified when the generation of porosities ceases and the keyhole remains relatively stable without significant fluctuations. It is worth noting that

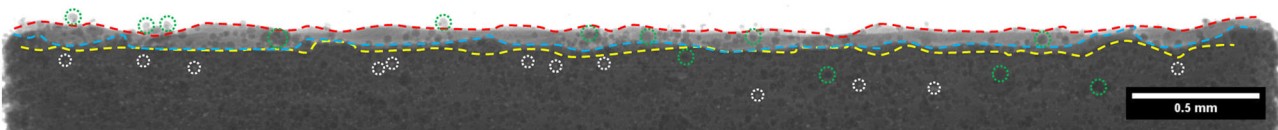

**Fig. 1 | An X-ray image of a thin 316 L stainless steel wall fabricated via the mini-LPBF device in as-built condition.** Multiple as-built surface profiles are highlighted with colored dashed lines. Lighter regions correspond to the presence of less material (lower thickness). A few of the partially attached metal powders to the top and side surfaces are encircled with green dashed lines. Some examples of keyhole porosities accumulated from the building process are encircled with white dashed lines. The full height of the wall is not presented in this image.

the time resolution of one X-ray frame (100 μs) introduces a potential uncertainty when precisely indexing the stable and unstable keyhole regimes. This uncertainty is associated with the duration of one X-ray frame and may affect the exact determination of the transition between these regimes.

Figure 2 exhibits the RM2 scanning condition (c.f. Supplementary Table 1) using an out-of-focus divergence beam (1.7, c.f. Supplementary Figure 1), leading to multiple instabilities from conduction to stable and unstable keyhole regimes. Melting starts in conduction mode (see Fig. 2a), lasting for about 20 ms, after which a short-lived transition to stable keyhole ($t = 20$ ms) and back to conduction ($t = 21.1$ ms) takes place, which is then followed by another transition to unstable keyhole regime ($t = 23.4$ ms). The generation and growth of the keyhole depression bring about enhanced laser energy absorption via the activation of multiple reflections[48]. Soon the melt pool starts to grow due to the drilling effect of the laser, with the melt pool depth

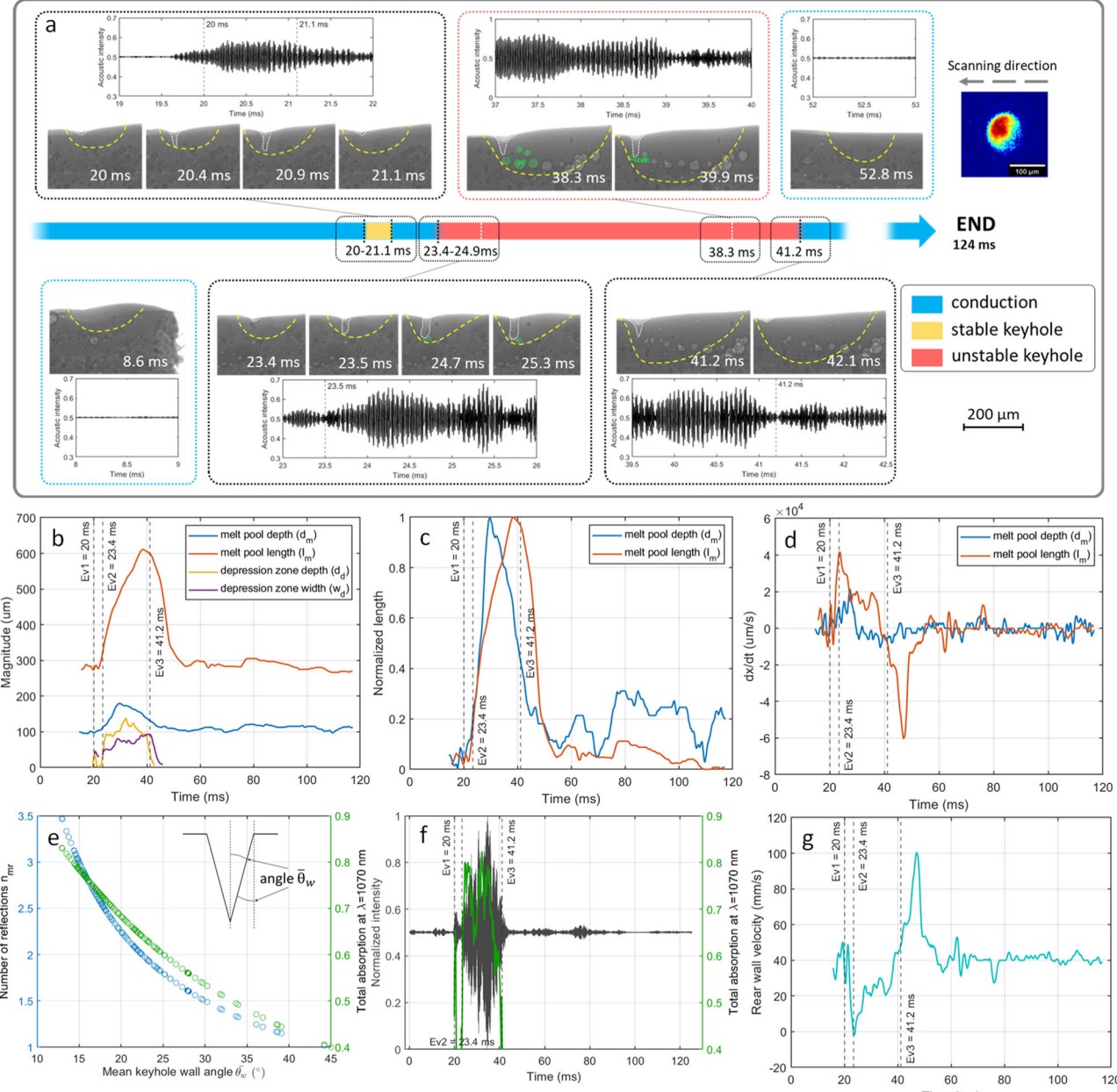

**Fig. 2 | Stochastic regime instabilities and subsequent transition from conduction to stable and unstable keyhole regimes. a** RM2 melt pool morphology variations under constant laser process parameters in a single scanning vector of 316L stainless steel (c.f. Supplementary Movie 2). The normalized and filtered acoustic signal for each time frame is illustrated for each image set with event annotations represented by vertical lines. The boundaries of solid-liquid (melt pool) and gas–liquid (depression zone) are illustrated by yellow and white dashed lines, respectively. The time bar depicts the regime changes from conduction to stable keyhole (Ev1, $t = 20$ ms), stable keyhole back to conduction ($t = 21.1$ ms), conduction to unstable keyhole (Ev2, $t = 23.4$ ms), and unstable keyhole back to conduction (Ev3, $t = 41.2$ ms). The resultant pores in the unstable keyhole regime in the vicinity of each timeline are encircled by dashed green lines. The scale bar for all the images is shown in the bottom right corner. The laser intensity profile (1.7, c.f. Supplementary Fig. 1) and the corresponding scanning direction are presented in the top right corner. **b** Melt pool length, depth, keyhole depth, and widths at the top of the depression throughout the scanning vector. **c** The normalized magnitude of melt pool length and depth over the scanning vector. **d** The first derivative of the melt pool length and depth magnitude over time. **e** Relations between mean keyhole wall angle and the number of laser reflections and total energy absorption. **f** Superimposition of total energy absorption evolution and the normalized and filtered acoustic signal (band-pass: 35–105 kHz). **g** Melt pool rear wall velocity measured via summation of laser velocity and melt pool length variations in time.

reaching its maximum ($d_m = 180\,\mu m$) at $t = 29.4$ ms (see Fig. 2b, c). At this point, the very fast regime transition, and the overheating at the bottom of the keyhole depression lead to an increase in the temperature gradient between the center of the depression and the boundaries of the melt pool, resulting in a pronounced Marangoni convection, evacuating, and redistributing the heat in the melt pool[15]. This is followed by shrinkage in the melt pool depth $d_m$, while its length $l_m$ continues to grow as a consequence of heat redistribution by Marangoni convection. The projected melt pool area was measured to be quite stable between $t = 29.4$ ms (where $d_m$ starts to decline) and $t = 38.6$ ms (where $l_m$ is at its maximum value), indicating the redistribution of heat in the melt pool to be the sole responsible for these changes. Additionally, we observed the formation of smaller porosities at both ends of the unstable keyhole regime duration ($t = 24.7$ ms and $t = 39.9$ ms) and larger pores in the middle of the track as a result of pore coalescence. This could be attributed to the coalescence of the smaller pores as their quantity increases at the beginning of the unstable keyhole regime. Meanwhile, at the final stages of the unstable keyhole, due to the drop in the absorbed energy and the following Marangoni convection magnitude, the melt flow dynamics decay resulting in a lower probability for the coalescence of smaller pores[7].

We used a simple welding analytical model describing the deep penetration of the laser, and considering the keyhole geometrical profile[49] to estimate the number of multiple reflections inside the keyhole cavity. This approach provides a first-order estimate of the time-resolved total energy absorption in the presence of a depression cavity. The number of reflections is estimated via $n_{mr} = \pi/4\bar{\theta}_w$, where, $\bar{\theta}_w$ is the mean keyhole wall angle, considering a limiting reflection angle of $\pi/2$. Subsequently, the total laser absorption including the first reflection is calculated using $\alpha_{tot} = 1 - (1 - \alpha_{Fr}) * (1 - \alpha_{mr})$, where, $\alpha_{Fr}$ is the Fresnel absorptivity and $\alpha_{mr}$ is the absorption coefficient for multiple reflections (for the detailed mathematical procedure, refer to Supplementary Method 1). Due to the complexity of the fluctuating keyhole wall morphology as a result of its exposure to a high-frequency pulsed laser beam, with polarizing reflectivity on a three-dimensionally curved surface, a more complete and accurate assessment of the absorptivity inside the cavity is not attempted. The simple modeling of absorptivity in the keyhole, solely based on the geometrical factor, is used here only as an indication on the came across correlation with the acoustic power (see Fig. 2e, f). A gradual change within the keyhole depression profile was observed at $t = 35.6$ ms, where the depression zone width $w_d$ grows and its depth $d_d$ decays, leading to a drop in the number of multiple reflections, and consequently in the total energy absorption. At this point both $d_m$ and $l_m$ decay until the transition to the conduction regime takes place. At that moment (Fig. 2b $-Ev3$, $t = 41.2$ ms), the reduced total energy absorption in the laser-material interaction zone[48] cannot provide the heat required to sustain the inherited melt volume, resulting in an increase in the velocity of the rear solid–liquid interface by a factor of two compared to the scanning velocity (speed of the front-end of the melt pool) (see Fig. 2d). This results in an accelerated speed in the movement of the solid/liquid boundary ($R$, $m/s$) at the rear part of the melt pool (Fig. 2g), and hence, in an increase of the cooling rate $G \cdot R$ ($K/s$), where $G$ ($K/m$) is the representative temperature gradient[50]. Accordingly, it can be said that the microstructural footprint corresponding to these regime instabilities can linger around after the disappearance of the unwanted instabilities. Due to the abrupt decrease in the amount of absorbed energy, the temperature gradient of the molten metal in the region close to the incident laser light and the melt pool boundaries drops rapidly. Consequently, a change in the dictating heat exchange mechanism from convection to conduction takes place[51], at distances close to the solid–liquid boundary, leading to a transition from keyhole to conduction regime.

The unstable keyhole was primarily attributed to the enhanced energy absorption via multiple laser reflections (up to $n_{mr} = 4$),

resulting in an estimated increase in total absorptivity as high as 0.85, more than twice the Fresnel absorption ($\alpha_{Fr} = 0.4$, at an incident angle of $\pi/2$) for 316 L stainless steel (see Fig. 2e)[7]. The variations in total laser absorption correlated well with the $l_m$ decay taking place at $t = 35$ ms and the drop in the AE signal amplitude (see Fig. 2f, g), suggesting the evaporation and formation of microjet as the principal factor in the generation of acoustic waves in keyhole melting regime[8]. This finding is in agreement with previous suggestions on the origin of acoustic waves in keyhole regime[8,39].

Figure 3 depicts the regime instabilities occurring under constant laser process parameters utilizing a convergent laser beam (−1.9, c.f. Supplementary Fig. 1) with a similar defocus distance as the divergent beam used in RM2 but having a dissimilar laser intensity distribution. RM6 (c.f. Supplementary Table 1) exhibited a continuous heat build-up in the laser-material interaction zone ensuing in a smooth transition from conduction to stable keyhole ($Ev1$, $t = 45.7$ ms) and eventually to unstable keyhole ($Ev2$, $t = 49.7$ ms). $Ev1$ is the onset of keyhole depression fluctuation and gradual growth (see Fig. 3a–c), with a melt pool expansion characterized by a depth growth velocity twice that of the melt pool length. At $t = 49.6$–$49.7$ ms, the fluctuation leads to a very sharp keyhole whose collapse results in the formation of porosity ($t = 49.8$ ms), followed by a new-born keyhole ($t = 50.1$ ms) and its transition to a needle-like keyhole ($t = 50.3$ ms) which, upon its growth and collapse leaves behind another porosity. This observation is in agreement with the keyhole porosity formation mechanism proposed in recent reports[8,39]. It has been demonstrated that the dynamic morphology of the front keyhole wall (FKW) brings about non-uniform laser absorption depending on the angle of incidence. In some occasions, these morphological features can shadow the bottom of the depression which could result in temperature and pressure drop followed by the collapse of the keyhole. The formed porosity may then go through pore migration under Marangoni-driven flow and potential pore coalescence[52,53] (see Fig. 3a, $t = 78.0$ ms). Furthermore, due to the very slow transition from conduction to stable and unstable keyhole regimes, no abrupt change in the velocity of the rear wall was observed (see Figs. 2g and 3g).

RM2 and RM6 (c.f. Supplementary Method 2) measurements were performed with similar beam sizes having an opposite defocusing position (positive and negative as divergent and convergent beams, respectively) and different laser intensity distributions. In conduction mode, the melt pool dimensions proved to be similar under identical laser processing parameters with a slight increase in melt pool depth and length in positive defocus, in agreement with previous observations[54,55]. Upon transition from conduction to keyhole, a significant difference in the melt pool length is observed with 45% higher values in RM2 compared to RM6. These melt pool length variations correspond directly to the local cooling rate upon solidification, leading to variations in the size of microstructural features[56]. At this stage, the defocus direction-dependent drilling effect and enhanced energy absorption becomes evident. As observed in Supplementary Figure 1, the convergent beam (negative defocus) possesses 28% higher energy densities in the very center of the beam as opposed to the divergent beam (positive defocus). Furthermore, the convergent beam generated a semi-conical shape depression zone (see Fig. 3a) with a quite wide entry on top and a very narrow keyhole bottom as a result of its laser energy intensity distribution and its converging inside the material, while, the divergent beam produced a narrow and deep cavity with slight variations in the depression width alongside the depth of the keyhole (see Fig. 2a). These observations were in accordance with the laser intensity profiles, with a concentrated intensity at the very center of the beam in the convergent beam and a more spread non-uniform energy distribution in the divergent beam (c.f. Supplementary Fig. 1f). Consequently, due to the lower aspect ratio, the number of laser multiple reflections decreased remarkably in the negative defocus, yielding lower total laser energy

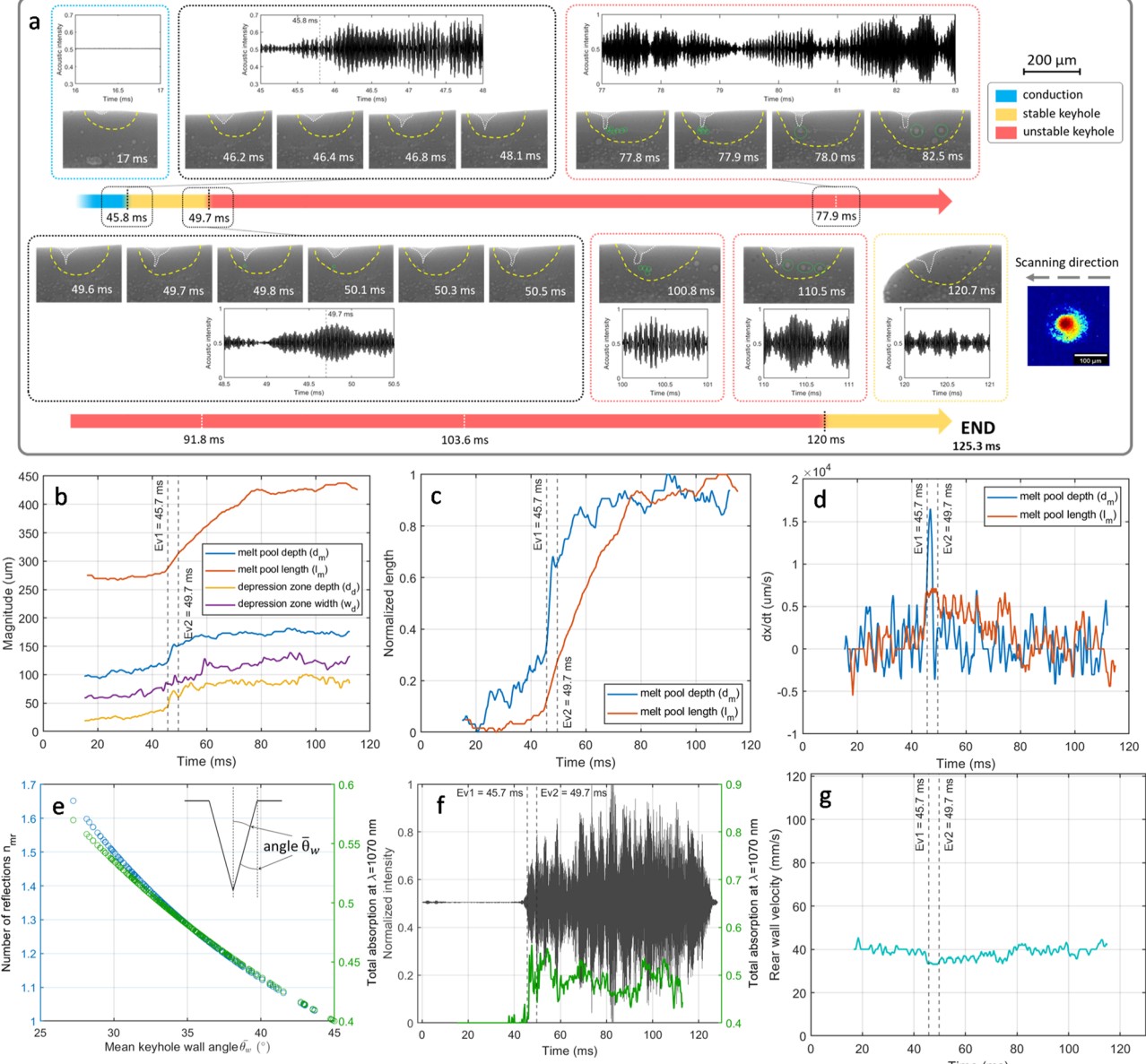

**Fig. 3 | Stochastic regime instabilities and subsequent transition from conduction to stable and unstable keyhole regimes. a** RM6 melt pool morphology variations under constant laser process parameters in a single scanning vector of 316 L stainless steel (c.f. Supplementary Movie 5). The normalized and filtered acoustic signal for each time frame is illustrated for each image set with event annotations represented by vertical lines. The boundaries of solid–liquid (melt pool) and gas–liquid (depression zone) are illustrated by yellow and white dashed lines, respectively. The time bar depicts the regime changes from conduction to stable keyhole (Ev1, $t = 45.7$ ms) and stable keyhole to unstable keyhole (Ev2, $t = 49.7$ ms). The resultant porosities in the unstable keyhole regime in the vicinity of each timeline are encircled by dashed green lines. The scale bar for all the images is shown in the top right corner. The laser intensity profile (−1.9, c.f. Supplementary Fig. 1) and the corresponding scanning direction are presented in the bottom right corner. **b** Melt pool length ($l_m$), depth ($d_m$), keyhole depth ($d_d$), and width ($w_d$) at the top of the depression throughout the scanning vector. **c** The normalized magnitude of melt pool length and depth over the scanning vector. **d** The first derivative of the melt pool length and depth magnitude over time. **e** Relations between mean keyhole wall angle the number of reflections, and total energy absorption. **f** Superimposition of total energy absorption evolution and the normalized and filtered acoustic signal (band pass: 35–105 kHz). **g** Melt pool rear wall velocity measured via summation of laser velocity and melt pool length variations in time.

absorption, almost by a factor of two. This observation was in good correlation with the difference in melt pool dimensions between the two RM2 and RM6 cases with distinct total laser absorption. The onset of a sharp increase in the AE signal was very well correlated to the formation of the depression zone while transitioning from conduction to keyhole, suggesting the evaporation and the formation of the metallic jet as principal actors in the generated acoustic signal while in keyhole melting mode. These observations were consistent through all experiments across different laser beams, defocus directions and nominal laser power/velocity combinations (c.f.

Supplementary Figs. 5–8 and the corresponding Supplementary Movies 1, 3, 4, and 6, respectively). Supplementary Figure 8 corresponds to two consecutive remelting passes, where the end of each laser scan was detectable via the acoustic signal. Furthermore, the effect of heat build-up as a result of the first scan is observed at the beginning of the second pass. In general, the conduction regime corresponded to AE amplitudes two orders of magnitude lower than those in the unstable keyhole regime. Generally, we observed an amplitude difference by an average factor of two between stable and unstable keyholes.

**Table 1 | Per-regime quality metrics obtained through a leave-one-out cross-validation evaluation scheme for all available signals**

| Metric | Ternary | | | Binary | |
|---|---|---|---|---|---|
| | Conduction [%] | Stable keyhole [%] | Unstable keyhole [%] | Conduction [%] | Keyhole [%] |
| Global F1-score | 97.04 | 69.90 | 89.83 | 95.89 | 98.34 |
| Global Precision | 97.39 | 58.32 | 96.81 | 95.44 | 98.53 |
| Global Recall | 96.70 | 87.21 | 83.79 | 96.36 | 98.15 |

The metrics shown include F1-score, precision, and recall. The F1-score, also known as the harmonic mean of precision and recall, balances the trade-off between the two and provides a comprehensive measure of overall performance. Precision, on the other hand, measures the proportion of true positive predictions among all positive predictions, while recall measures the proportion of true positive predictions among all actual positive instances. In addition to the per-regime metrics, the table also includes the global figures, calculated by concatenating all the obtained predictions and evaluating them against the concatenated ground truths. These global figures provide a comprehensive overview of the overall performance of the model across all regimes.

## Segmentation results

In this study, we introduce an approach for the segmentation of AE signals in conduction, stable, and unstable keyhole regimes. The aim is to precisely identify and divide the segments, or signal portions, corresponding to each of these regimes. To evaluate the ability to accomplish this objective, we employed a leave-one-out cross-validation evaluation scheme. This approach enables efficient use of the available data and provides significant predictions for each signal, allowing us to robustly and accurately evaluate the performance[57]. Specifically, the used optimization procedure, detailed in Supplementary Method 3, is reinitiated as many times as the number of available signals. Every time the optimization procedure is restarted, we exclude one of the available signals from the parameter update, which includes the gradient computations of the cumulative risk over the remaining signals. This methodology allows for a robust and accurate evaluation of the performance of the excluded signal.

In order to evaluate the quality of the segmentation obtained by the algorithm, we employed the F1-score metric. This measure, commonly used in machine learning and information retrieval, provides a comprehensive evaluation of the algorithm's performance by considering both precision and recall and is calculated as the harmonic mean of these two metrics. To be specific, precision is defined as the proportion of correctly predicted positive instances or data points predicted as belonging to the correct regime, out of all data points predicted as belonging to that specific regime. On the other hand, recall is the proportion of correctly predicted positive instances out of all actual data points in that regime. This metric assesses the algorithm's performance in terms of its ability to correctly identify segments and predict their belonging to a specific regime.

The results of this study are presented in Table 1, where several quality metrics per regime are reported using the leave-one-out cross-validation evaluation scheme for all available signals for both ternary (conduction, stable, and unstable keyholes) and binary (conduction and keyhole) segmentation problems. As can be seen, the algorithm segments all signals correctly, particularly for the conduction regime (for both binary and ternary problems), keyhole (for the binary case), and unstable keyhole (for the ternary scenario)—allowing for the detection of unwanted porosities or defects in the manufactured parts.

More specifically, if we analyze the quality metrics per regime, we can see that in both cases (binary and ternary), all figures of merit are very high (>95%) for the conduction regime. The ease of conduction detection can be attributed to the segments identified in the X-ray movies as corresponding to the conduction regime, which present little frequency content throughout the acquisition band (10 Hz–1 MHz) and correspondingly low signal amplitude.

This consideration also makes keyhole detection in the binary case a straightforward problem by exclusion. However, the same cannot be said for the ternary problem, particularly for stable and unstable keyhole detection. In this situation, as seen from Table 1, being able to discriminate between stable and unstable keyhole is a challenging task, as the F1-score drops to about 70% for stable keyhole, whereas it stands at almost 90% for the unstable keyhole.

To investigate the causes of this behavior, we can look at the other two available figures of merit in Table 1, precision and recall. For stable keyhole, the origin of the lower F1-score is a reduced regime-specific precision compared to recall, i.e., most segments corresponding to stable keyhole are correctly detected as such (thus, high recall), but not all segments recognized as stable keyhole are correct (thus, low precision). On the other hand, the high F1-score for conduction and unstable keyholes allows us to conclude that most errors are unstable keyhole segments mistaken for stable keyholes—and this is confirmed by the confusion matrix in Supplementary Fig. 9, which provides a visual representation of the percentual number of correct and incorrect predictions made by the algorithm.

Additionally, Fig. 4 shows the output obtained for two of the available signals, the algorithm predictions, and the corresponding ground truth. The optimized filters can discern between the different regimes by producing signals with a high value when the corresponding regime is occurring and a low one when the other regimes are detected. Furthermore, most of the model errors are located close to the regime transition instants. This is due to the fact that the ground truth was derived from X-ray movies, which have a lower temporal resolution compared to acoustic emission. Specifically, the X-ray movies have a sampling frequency of 10 kHz, resulting in a time distance between frames of 0.1 ms, while acoustic emission is sampled at 2 MHz, resulting in a time resolution of 0.5 µs. Therefore, the effective accuracy of the model is considered to be higher than what is reported.

Moreover, Fig. 5 shows the frequency response of the filters before and after optimization. Specifically, in both binary and ternary cases, the frequency response of the conduction filter shows a precise amplification of the frequency contents in the lower end of the spectrum and a distinct peak centered at 250 kHz—without significant differences compared to the initial conditions (apart from the overall reduction of the in-band attenuation). This peak at the 250 kHz pulse repetition rate may correspond to the periodic evaporation at the surface of the melt pool provoked temporally by the laser pulse repetition rate, taking place at the center of the laser profile, where the laser intensity is at its maximum. However, the low-frequency content is due to being in the conduction regime, which means that due to high reflectivity accompanied by low energy absorption and low evaporation rate, the evaporation-induced recoil pressure is insufficient to overcome the magnitude of surface tension for creating a depression cavity[15,16], and, thus, the melt pool dynamics are much slower compared to the keyhole regime.

On the other hand, for the binary case, the keyhole filter is significantly changed after optimization compared to its initial conditions—showing the amplification of the energy content limited to the frequency band 40–80 kHz and around two higher frequency peaks, centered at 250 and 500 kHz, respectively. These two higher spectral peaks correspond to the fundamental and second harmonic

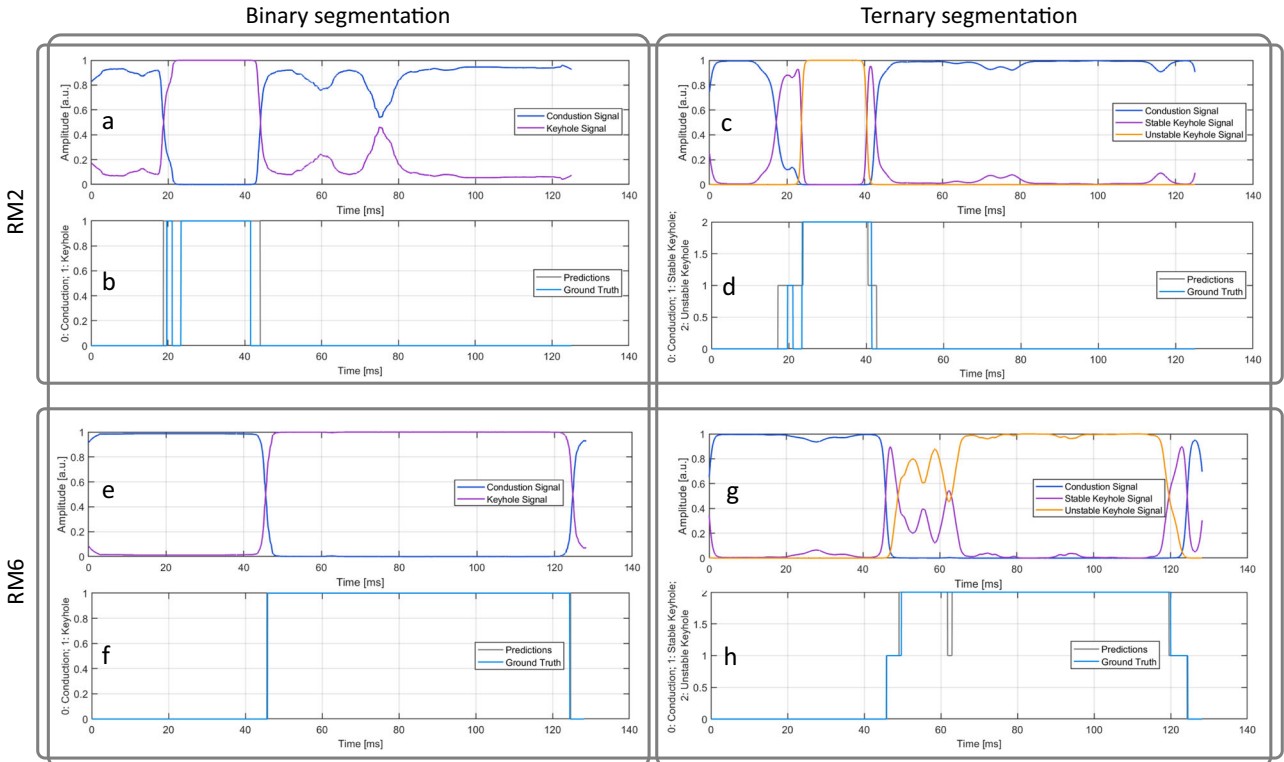

**Fig. 4 | Time evolution of segmentation algorithm output for RM2 and RM6 signals.** Time evolution of the output of the segmentation algorithm (after SoftMax normalization) obtained for two signals (RM2 (**a**, **c**) and RM6 (**e**, **g**)) for both the binary (**a**, **e**) and ternary (**c**, **g**) segmentation problems. **b**, **d**, **f**, **h**) In blue (gray), the time evolution of the ground truth (predictions obtained) is presented. As can be seen, the optimized filters can discern between the regimes by producing signals with a high value when the corresponding regime is occurring and a low one when the other regime is detected. Results from the leave-one-out cross-validation evaluation scheme, i.e., RM2 and RM6 were not used to optimize the filters' parameters.

of the laser pulse repetition rate—as described in Supplementary Method 3—and may be related to the laser pulse-induced evaporation at the bottom of the keyhole cavity, as formulated by[58]. Additionally, it is possible to recognize five spectral peaks amplified by the filter, located at 40, 60, 70, 77, and 82 kHz, respectively. These findings match what has been observed in[59], where the signal energy content in the 40–90 kHz frequency band is correlated to the welding penetration depth, and 60 kHz is identified as the primary frequency component during keyhole welding for 316 L stainless steel and Ti6Al4V alloy.

Turning to the ternary segmentation problem, Fig. 5 illustrates how the filter for unstable keyholes exhibits less attenuation in the lower frequency spectrum compared to the stable keyhole filter. The main distinction between the two filters can be observed in the absence of the 250 and 500 kHz peaks for the unstable keyholes. Despite this, the five previously mentioned spectral peaks are still present in both stable and unstable keyhole filters. This is a result of the more unpredictable dynamics of the laser-material interaction in the unstable keyhole regime, which leads to the repetitive collapse of the keyhole depression cavity, disrupting the periodicity of the laser pulse-induced evaporation at the bottom of the keyhole cavity. This disturbance can be attributed to various factors, such as variations in the keyhole front wall morphology, which can cause shadows on the bottom of the keyhole, leading to temperature and pressure drops in the lower keyhole depression cavity, ultimately causing the collapse of the keyhole[8,12].

## Discussion

In this study, we present a comprehensive analysis of inter-regime instabilities and the utilization of AE for fast and accurate detection of regime transitions during laser melting processes. To this aim, we investigated the following transitions: from conduction to stable keyhole and from stable to unstable keyhole regimes and vice versa—all observed in individual line scans under different sets of constant processing conditions. These experiments were validated throughout a range of beam shapes and sizes, laser defocus directions and distance, and laser scanning parameters. Our findings reveal that these instabilities result in transitions from steady conduction melting mode to the stable keyhole and eventually to unstable keyhole with a recurring collapse of the keyhole depression cavity, leaving behind a sequence of spherical pores that are eventually pinned by the moving solid-liquid boundary. Furthermore, the detrimental effects of these unwanted transitions are not limited to the formation of porosities but include also the microstructural imprint[60], which lingers even after transitioning back to conduction mode due to variations in the solidification velocity at the solid–liquid interface.

To detect the abovementioned instabilities, we collected the AE signals emitted from the remelting experiments of an LPBF-produced thin wall and successfully correlated them to the corresponding regime transitions determined by the operando X-ray imaging data, resulting in a segmentation technique for regime prediction with accurate time resolution. Notably, the acoustic footprint of these instabilities showed not to be affected by changes in beam and laser scanning parameters; hence, they could be identified as universal regime-related acoustic signatures for the studied material. Moreover, the amplitude of the AE correlated well with the shape of the depression cavity[49,61]—suggesting that the solid-gas phase transformation in the laser-material interaction zone and the subsequent microjet formation[8] are the principal factors responsible for higher acoustic pressure measured in the keyhole mode. These findings elaborated on the factors responsible for the generation of acoustic waves in the LPBF process.

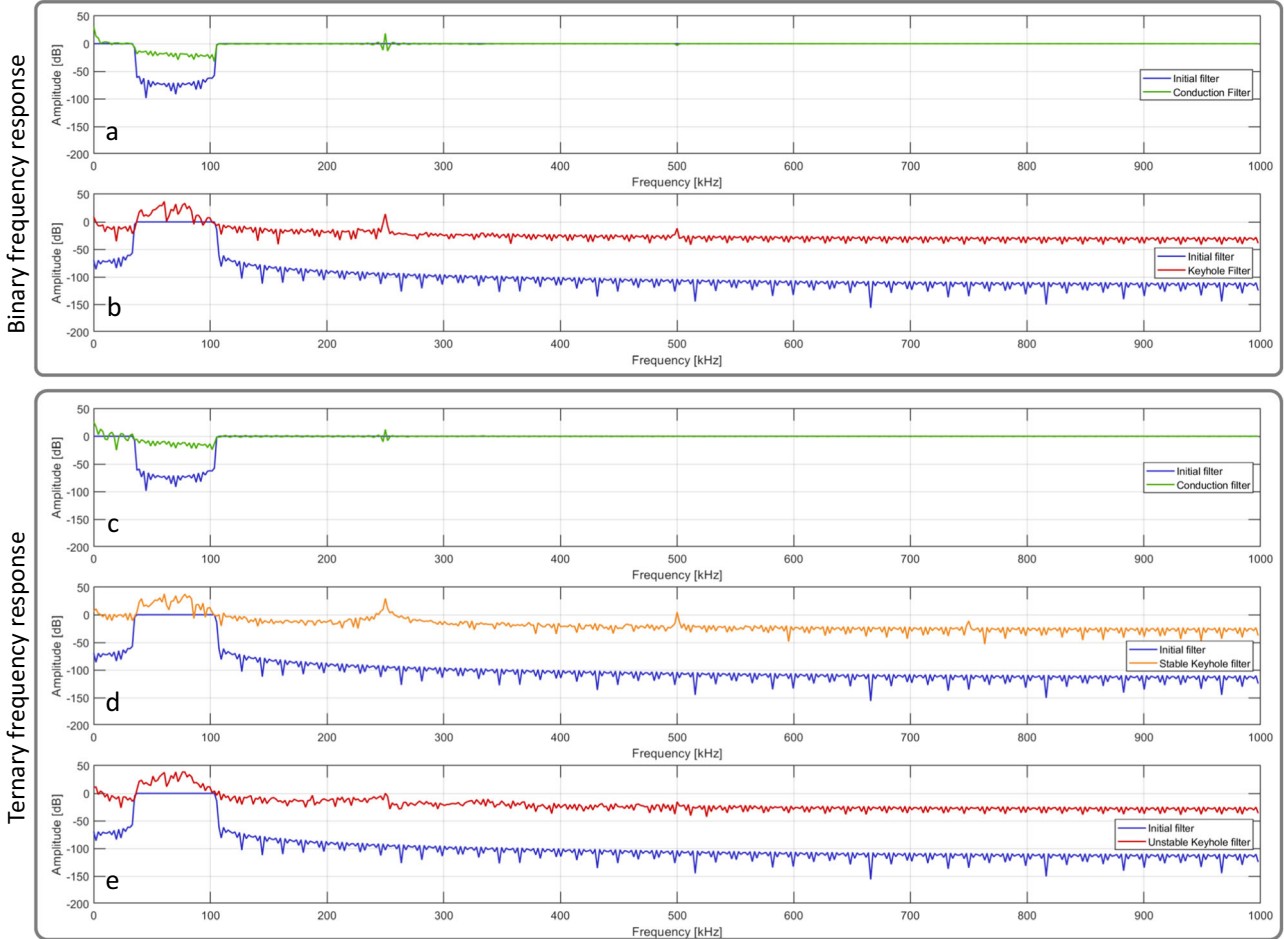

**Fig. 5 | Filters frequency response before and after optimization for both the binary (top box) and ternary (bottom box) segmentation problems.** In the ternary case, the top plot (**c**) in green illustrates the frequency response of the conduction filter, which shows a distinct amplification of the frequency contents at the lower end of the spectrum and at the frequency corresponding to the pulse repetition rate (250 kHz). In contrast, the center plot (**d**) in orange and the bottom plot (**e**) in red show the frequency response of the band-pass filters for stable keyhole and unstable keyhole, respectively. The filter for the unstable keyhole has a lower attenuation at the lower end of the spectrum compared to the stable keyhole filter. However, the key difference between the two is the absence of the 250 kHz peak in the unstable keyhole filter. Similarly, for the binary case (top box), the top plot (**a**) in green illustrates the frequency response of the conduction filter, which again shows a specific amplification of the frequency contents at the lower end of the spectrum. In contrast, the bottom plot (**b**) in red shows the keyhole filter compared to the initial filter.

Although the X-ray experiments were conducted in the absence of powder on already built thin walls via LPBF, recent studies have reported no significant difference in keyhole depth in the presence of powder bed[8,12]. As a matter of fact, the keyhole threshold in process-parametric space only shifts marginally from bare plate to powder bed setup[8]. Moreover, the regime instabilities are expected to be even more recurrent in the presence of powder due to effects such as variations in the powder bed thickness[62], powder particle entrainment and laser shadowing[17], and more pronounced variations in the laser absorption alongside the scanning line[17].

This study provides an approach for detecting the stochastic regime instabilities in the LPBF process via real-time/in-situ monitoring utilizing the AE data recorded from the laser-material interaction zone. By treating the raw AE data via a segmentation algorithm consisting of several adaptive filtering branches, we detected regime transitions with high accuracy. Specifically, the processing pipeline had been designed in such a way as to be flexible enough to be tuned by employing gradient descent using the collected data while limiting the parameters optimization to the filters' impulse response. Furthermore, this architecture facilitates the interpretation of the results in terms of filtered or amplified frequency content—enabling the identification of six spectral peaks primarily responsible for adequate signal segmentation. Specifically, the peaks at 40, 60, 70, 77, and 82 kHz enable discrimination between conduction and keyhole modes, while the laser pulse repetition frequency peak at 250 kHz is essential for discerning stable from unstable keyhole regimes. The use of a pulsed laser in LPBF offers benefits for the detection of unstable keyholes, as the associated erratic melting dynamics disrupt the periodicity of the laser pulse-induced evaporation at the bottom of the keyhole cavity, allowing for instability detection through the monitoring of acoustic energy content in proximity to the pulse repetition frequency. This capability is of significant importance for monitoring LPBF processing, especially for detecting instabilities within the keyhole regime, where porosity formation occurs. Furthermore, with high time resolution in these predictions, the location of affected regions can be accurately identified, and remedial measures can be employed to save processing time and resources. With these findings, we cannot only improve the efficiency of the LPBF process but also ensure the high quality of the end product.

## Methods

### LPBF apparatus and X-ray imaging
The experiments were performed on LPBF printed walls fabricated from a gas-atomized 316 L stainless steel (1.4404) powder (OC

Oerlikon Corporation AG, Switzerland) (c.f. Supplementary Method 4). A miniaturized LPBF device designed at the Paul Scherrer Institute (PSI)[63] was chosen for this study. This setup mimics the commercial LPBF process and can be incorporated into synchrotron beamlines for in situ X-ray measurements simultaneously (c.f. Supplementary Method 5). The apparatus is equipped with two glassy carbon windows allowing the X-ray beam to access the powder bed from the rear window and the transmitted X-ray beam to reach the detector placed outside of the chamber through the front window. A dual-mode fiber laser (redPOWER, SPI lasers Ltd., UK) with a maximum power of 500 W operating at a wavelength of $1070 \pm 10$ nm was used. The laser was utilized in pulsed mode with a pulse frequency of 250 kHz and a nominal laser pulse duration of 2 μs.

The operando X-ray imaging experiments were carried out at the TOmographic Microscopy and Coherent rAdiology experimenTs (TOMCAT) beamline of the Swiss Light Source (SLS) using the LPBF apparatus described earlier. The polychromatic X-ray spectrum, generated by the 2.9 T bending magnet and filtered with 5 mm of Sigradur and 200 μm of Si was utilized for the experiments providing energies ranging between 10 and 55 keV. The peak of the energy spectrum was around 20 keV. The X-rays were allowed to pass through the 316 L stainless steel walls while the laser was impinging perpendicularly onto the top surface of the wall. Consequently, the propagated X-rays were converted into visible light using a 150 μm thick LuAg:Ce scintillator and recorded with a 4× microscope (Optique Peter, France)[64], coupled to an in-house developed GigaFRoST detector[65]. This detector exhibits a 2016 × 2016 pixels CMOS imaging chip with an 11 μm pixel size and 12-bit nominal dynamic range. In addition, its readout system provides continuous and sustained data streaming up to almost 8 GB/s to a dedicated high-performance data backend server. The imaging was carried out with a frame rate of 10 kHz, a pixel size of 2.75 μm, and an exposure time of 95 μs for a single X-ray image. This performance was achieved by reducing the region of interest (ROI) to 2016 × 200 pixels, equivalent to a field of view of 5.54 × 0.55 mm². The X-ray imaging was triggered by a TTL signal provided by the laser control card to ensure full synchronization between the start of the laser and X-ray acquisition.

## Image processing pipeline

The X-ray movies obtained during the laser processing were analyzed using the ImageJ[66] software package. Each image stack was examined frame by frame, providing the ground truth for variations in the melt pool geometry and alterations in the melting regime between conduction, stable keyhole, and unstable keyhole. Conduction and keyhole regimes are distinguished via the presence of a depression zone, with an unstable keyhole considered as the keyhole with a repetitive depression zone collapse resulting in the formation of keyhole pores[12]. An image-based quantification approach was employed on the acquired X-ray radiographs $(x, y, t)$ to measure the melt pool and depression zone geometry. Visual analysis reveals that a slight change in the gray levels occurs when the material is molten, the detection of which gives the maximum depth and lateral extension in the imaged plane. In an ideal case (for measurement reproducibility), a fully automated detection of the melt pool limits based on the acquired images would be implemented. However, the low signal-to-noise ratio (SNR) and complex evolutions that are visible—melt pool evolution and keyholing, thermal strains, porosity formation, and entrapment—turn this task into a significant image analysis challenge, making a conventional active contours approach[67] unsuitable for the identification of the edges of the melt pool due to the low stability throughout the image set. Furthermore, due to the low SNR, it proved difficult to pick out the melt pool edges in the visual inspection of a single radiograph. However, its edges can be identified in its temporal context (i.e., when played as a movie). The developed melt pool quantification pipeline was split into multiple steps: time-windowing,

denoising, detection of a keyhole and melt pool edges on the melt pool surface, and annotation of the melt pool boundaries and keyhole depression.

**Time-windowing.** Exploiting the fact that the radiographs are acquired at a constant frame rate and that the laser moves at a constant speed, the time series of radiographs has a moving window or ROI extracted around the laser's position in each frame. The laser speed in the coordinate system of the radiographs is measured with a line fitted in the $x–t$ domain using support vector regression. The result is a time series where the laser appears stationary and the texture of the sample "flows" past.

**Denoising.** A median filter (7 × 3 × 3 footprint where the first dimension is time and the other two are pixels) is applied to denoise the time-windowed image. The median filter aims at strengthening the signal of the relatively stationary melt pool against the flowing texture of the sample. Furthermore, the constant-speed movement of the texture creates diagonal lines with a fixed slope, which are removed to some extent with a stripe removal algorithm[68].

**Detection of the melt pool and keyhole depression edges at the melt pool surface.** The detection of the front and rear boundaries of the melt pool and keyhole (when present) requires a definition of the top surface of the sample since it varies both because of the initial configuration of the sample and due to the effect of the laser. Given the very strong contrast between the sample and the non-attenuating top surface, they are easily separated with an automatic Otsu threshold[69]. The pixels at the edge are the surface pixels, which follow the low-spatial-frequency variations in the surface of the sample. To better detect the almost-vertical edge of the melt pool and the keyhole depression (when present) at the top surface, the point where the laser touches the top surface in each image is used as the origin of a polar coordinate system. Radial image gradients are calculated for all pixels, detecting radial changes of gray levels, notably the edge of the melt pool. The radial gradient image is averaged vertically over 5 pixels from the identified surface, obtaining a horizontal line average for each time step, which assembled makes a 2D image with one axis being time and the other the surface length in the ROI. This spatiotemporal image (see Supplementary Fig. 10) shows lines that are slowly changing with time and that correspond to the edges of the melt pool and depression zone, respectively; these are manually annotated with an appropriate tool and converted back into the global coordinate system to yield front and back positions along the identified moving surface for the melt pool and keyhole (where present) for each timestep (0.1 ms).

**Depth annotation.** The measurements of the depth of the melt pool and possible keyhole depression were difficult to automate; thus, it was decided to implement a live graphical annotation tool (in the form of a plugin for the Python image viewer Napari[70]) to allow the time series to play at a reduced frame rate (10 FPS), allowing for tracking the point of interest (i.e., the bottom of the melt pool and potential keyhole depression) with the mouse, the position of which is recorded automatically frame-by-frame. Furthermore, this annotation can be converted back to the global coordinate system. Output from image analysis and annotation, therefore, are time series of points featuring the melt pool and the depression zone (i.e., melt pool length ($l_m$), melt pool depth ($d_m$), depression depth ($d_d$), and depression width ($w_d$)) (see Supplementary Fig. 11 in addition to the earlier Fig. 2 and Fig. 3).

## Acoustic measurement and signal processing pipeline
**Acoustic emission acquisition.** A membrane-free optical microphone (Eta250 Ultra, XARION Laser Products, Austria) was selected as an acoustic sensor to record airborne acoustic emissions from the laser-material interaction zone conveyed through the inert atmosphere. The

microphone uses the principles of interferometry to collect a signal whose frequency range extends from 10 Hz to 1 MHz. Precisely, the core of the microphone consists of an optical interferometer made of two semi-transmissive mirrors arranged at a distance equal to a multiple of its laser half wavelength to induce constructive interference of the transmitted laser beam. Sound waves passing through the microphone's etalon are measured via their induced density modulation of air and the resultant alterations in the optical refractive index of the medium, which in turn influences the laser propagation speed and wavelength passing through the etalon. Consequently, the distance between the two mirrors no longer satisfies the condition for constructive interference. By employing interferometry, laser intensity variation and the pressure level representing the sound wave can be estimated[71,72]. The XARION microphone was placed in the LPBF chamber at 100 mm from the powder bed right on top of the processing zone with its mirrors parallel to the build plate (see Fig. 1b) to maximize the recipient information from the laser-material interaction zone[72].

**Segmentation algorithm.** Our approach to segmenting the melting regimes for LPBF processes is built upon initial empirical observations of the available data, specifically the time-frequency analysis of the signals. Through this examination, we were able to identify specific spectral characteristics that correspond to the conduction and keyhole regimes. To refine our findings, we first applied signal pre-processing techniques. As the acoustic data acquisition was manually triggered shortly before the start of the laser irradiation, we exploited the high signal-to-noise ratio (SNR) to remove sections of the signal where the laser is off by simple thresholding. Upon closer examination of the acoustic amplitude changes at the beginning and end of the laser scan, it was observed that there is a region of uncertainty spanning approximately 100 μs. It is important to note that this uncertainty aligns with the time resolution of the X-ray imaging, which indicates that the precise indexing of the stable and unstable keyhole regimes may have an associated uncertainty of approximately 100 μs. The signals are then normalized using the RobustScaler algorithm (c.f. Supplementary Method 6).

After pre-processing, we conducted exploratory data analysis (EDA) on the pre-processed signal spectrograms. Our observations showed that the majority of the signal energy was concentrated in the

frequency range of 35–105 kHz when the process was in the keyhole regime—c.f. (Fig. 6). To improve the pipeline's expressivity, we applied a digital filter to extract only the relevant frequency range. Additionally, we enhanced the filtered signal by applying a non-linear function point-wise; the selected non-linear function is PReLU[73] which facilitates the application of the successive smoothing filters by making the signal running average positive.

Furthermore, we duplicated the filtering pipeline to produce two signals, each corresponding to a specific regime of interest. The implementation of two filtering branches offers several benefits, including increased robustness, a framework that can be easily expanded to detect multiple regimes, and the ability to discriminate between the conduction and keyhole regimes by applying the SoftMax function (c.f. Supplementary Method 3).

However, we recognized that relying solely on these observations may not deliver the most optimal results for all available signals. Therefore, to optimize the discrimination of the regimes, we adopted a data-driven approach to improve the pipeline further. This approach utilizes annotated data to fine-tune the pipeline, going beyond the initial time-frequency insights and applying a data-driven filter that allows one of the two output signals to be more intense when the corresponding regime is occurring. This fine-tuning allows for a more accurate and reliable segmentation of the regimes, which is crucial for real-time/in-situ monitoring of the LPBF/Laser melting process. Specifically, to guide the filter design optimization, we used a variation of the cross-entropy loss, which ensures a low score when the pipeline segmentation matches the ground truth and a high score otherwise. The ground truth, in this case, refers to the annotated data via X-ray movies that provide the true labels for each data point, indicating whether it corresponds to the conduction or keyhole regime. The pipeline's parameters are updated using the gradient descent technique[74], with the gradient calculations performed using an automatic differentiation tool such as Pytorch[75] (see Supplementary Method 3).

**Unstable keyhole detection.** In the previous section, we presented a regime segmentation technique for the LPBF/Laser melting process based on the analysis of AE signals. We used a filtering pipeline to extract specific frequency ranges and a prediction model utilizing the SoftMax function to discriminate between the conduction and keyhole

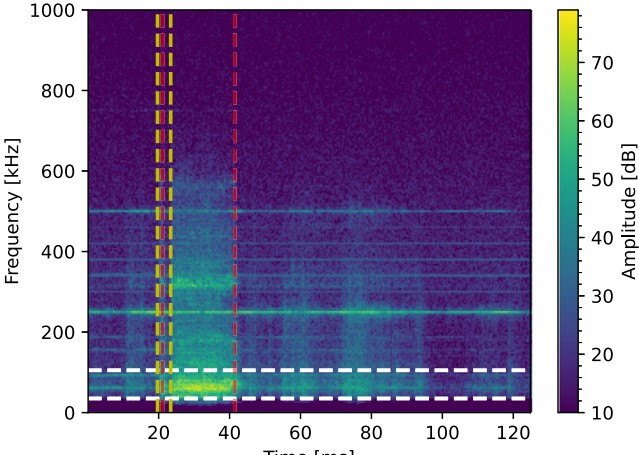

**Fig. 6 | Time evolution of one AE signal (RM2) annotated based on the X-ray movies (on the left) and its spectrogram representing the signal power spectral density (on the right).** The orange signal on the left corresponds to the ground truth, which has a value of 1 when the keyhole regime is occurring and 0 when the process regime is conduction. The vertical yellow (red) dotted lines on the right plot represent the start (end) of the keyhole regime. As can be noticed, most of the

signal energy content is located in the frequency range 35–105 kHz when the corresponding regime is keyhole (region highlighted by the horizontal dotted white lines). To obtain the spectrogram, the AE signal is split into segments of 1024 data points each, with 512 overlapping data points between consecutive segments. The Hanning windowing function window is then applied to each segment, and the spectrum of each section is computed.

regimes. However, to fully utilize the information provided by the AE signals, we further optimized the technique by allowing for the discrimination of the unstable keyhole regime from the stable one. This accomplishment was achieved through minor modifications to the pipeline, such as adding an additional filtering branch and slight changes to the loss function. For more information, please refer to the Supplementary Method 3. The extended capability is crucial for LPBF/Laser melting processing monitoring, specifically detecting instabilities within the keyhole regime and identifying affected regions for timely healing measures to save processing time and resources.

## Data availability
The representative data that support the findings of this study are presented in the figures and tables within the main paper, Supplementary Information, and corresponding movies. Additional datasets that underlie the results of this research are available upon request from the corresponding author.

## Code availability
The code employed for image processing in this study can be accessed and found at the following public repository: https://gitlab.com/epfl-center-for-imaging/sti_lmtm_melt_pool_tracking[76]. The code employed for the implementation of the segmentation algorithm in this study can be accessed and found at the following public repository: https://github.com/GiulioMa/Adaptive-Filter-for-Acoustic-Signal-Segmentation/tree/main[77]. Researchers interested in replicating or building upon the methods used in this study are encouraged to refer to these repositories for access to the relevant codes.

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

## Acknowledgements

The authors are grateful to Dr. Edward Ando and Mallory Wittwer at EPFL center for imaging for their help in image analysis of the acquired movies, and Dr. Toni Ivas for his assistance with empirical models for laser absorptivity in keyhole melting regime. We acknowledge the Paul Scherrer Institut, Villigen, Switzerland for the provision of synchrotron radiation beamtime at the TOMCAT beamline X02DA of the SLS. The authors would also like to acknowledge the support from Mr. Ryan Sommerhuber, Application Engineer, XARION Laser Acoustics GmbH during the set-up of the XARION sensor. M.H.N., Cd.F., S.V.P., G.M., R.E., K.W., V.P. and R.E.L. acknowledge funding from the Swiss NSF Sinergia project CRSII5_193799. M.H.N., S.V.P., L.S. and R.E.L. acknowledge funding from the SMARTAM project, within the Strategic Focus Area Advanced Manufacturing (SFA-AM) initiative of the ETH Board in Switzerland. All members of the Thermomechanical Metallurgy Laboratory at EPFL acknowledge the generous financial support of PX Group to the laboratory.

## Author contributions

M.H.N., R.E.L., S.V.P. and K.W. conceived the project. M.H.N. wrote the paper with contributions from G.M. and C.d.F. Characterization and image processing were performed by M.H.N., F.A., Cd.F, and L.S. Conceptualization and experimental work were led by M.H.N., with contributions from Cd.F., L.S. and R.E. Acoustic data analysis was conducted by G.M. and M.H.N. Acoustic system installation was carried out by L.S., V.P. and S.G., with valuable acoustic device support provided by A.G. and A.S. K.W. supervised the work at Empa, while F.M. and S.V.P. played crucial roles in the installation of the system at the beamline and the synchronization of the laser system and X-ray imaging. G.M. should be the point of contact for matters related to the segmentation algorithm. S.V.P. should be approached for queries regarding the X-ray imaging setup. For all other correspondence related to the article, please direct your inquiries to M.H.N., who is the lead researcher overseeing the broader aspects of this study.

## Competing interests

The authors declare no competing interests.
