## [Peer Review File · Nature Communications]

REVIEWER COMMENTS

Reviewer #1 (Remarks to the Author):

Increasing the understanding of the physics underpinning the LPBF process is critical to its greater adoption by the different industry sectors so I read your paper with much interest. I would like to commend the team on the design of experiments, their detail and depth and the comprehensive and detailed analysis of the results. I have a few comments that you may want to consider before the paper is published.

1. I believe the title needs to be more reflective of the reported work. Currently it is focused on the X-ray imaging but I would focus on AE.
2. Too much emphasis is placed on LPBF when in fact the work is all about laser melting and AE. I note that you have commented on the link between the two but what was not clear to me is whether you collected any AE signals during the manufacture of the SS wall or all the AE signals reported relate to the laser melting of the wall. If this is the case than the paper should just focus on the laser melting experiments and the connection to LPBF should not be that prominent.
3. It is not clear to me why is conduction mode observed when the chosen laser parameters, as mentioned in the paper, were meant to only generate the keyhole mode. I think further clarification on this would be helpful.
4. When repeating the laser melting scans for individual parameters was the top surface of the wall machined between the scans. The roughness of this surface can significantly affect the absorptivity.
5. I suggest changing the laser power to average laser power throughout the text.
6. Can the acoustic signal graphs be made larger. I found them difficult to read in their current version.
7. Fig 3f has a laser wavelength of 1064nm but the rest of the paper refers to 1070 nm.
8. Fig 3f also shows that there appears to be lack of close overlap between the duration of the AE signal and the total absorption.
9. Table 1. There appears to be misalignment between column headings and columns.

M. Brandt

Reviewer #2 (Remarks to the Author):

This manuscript presents work that is interesting, but is not well suited to the broad audience of Nature Communications and should instead be published in a more specialized journal focused on additive manufacturing. The authors describe experiments that apply operando synchrotron X-ray imaging and acoustic emission monitoring simultaneously to the laser powder bed fusion (LPBF) process. Based on the acoustic monitoring, they train a data-driven model to categorize the LPBF process into 3 different processing regimes: conduction mode, stable keyhole mode, and unstable keyhole mode. They claim that this is the first report of "operando synchrotron X-ray imaging synchronized with acoustic emission recording". If this claim were true, then this work might merit publication in Nature Communications. Unfortunately, it is not. Pandiyan et al (Additive Manufacturing, 2022, 58, 103007) published a previous report on a very similar topic, in which they use operando synchrotron X-ray imaging collected simultaneously with acoustic sensing and other sensing modalities to categorize the laser powder bed fusion process into three process regimes: lack of fusion, conduction mode, and keyhole mode. Due to this extremely similar prior publication, this work is better suited for a more specialized journal.

In addition to this oversight, multiple citations do not actually support the claims the authors make when citing them. These issues must be resolved prior to publication in any journal.

Specific comments:

Lines 72-87: this section is poorly written and it is not clear what point the authors intend to make. The authors list a variety of hypothetical mechanisms that might lead to acoustic emission during LPBF. But it is not clear if the authors are actually asserting that any of these mechanisms are relevant to their study, and this section would be improved if it were not written in a hypothetical tense. Furthermore, some of the literature the authors cite in this passage of text do not match the claims made. For example in line 72 the authors cite Martin et al (ref 30) to claim "noise is produced during the LPBF process, which can be employed as an in situ feedback signal". This reference does not report acoustic emission measurements or report any attempts at feedback during LPBF. In lines 74-78, the authors describe 3 potential mechanisms for acoustic emission during laser welding and cite references 32-34. Reference 33 describes shock dynamics and liquid explosions in water droplets caused by X-ray pulses, a substantially different physical process than laser welding. The authors should modify this section to ensure they are citing appropriate references.

Line 105: "for the first time" - as mentioned earlier, this statement is incorrect as Pandiyan et al (reference 27) reported very similar work in 2022.

line 111: "a data driven filtering pipeline" - it is unclear how the "data driven filtering" approach the authors mention here and describe in the supplementary information is substantially different than "machine learning" or "artificial intelligence" that the authors claim to be fundamentally flawed in the previous paragraph. While the specific architecture of their data driven model is different than the

previous work the authors disparage above, they still employ a training-based model that is susceptible to the same challenges many machine learning approaches face when differentiating between highly correlated signatures that arise from complicated physical system such as LPBF. A more nuanced discussion of what differentiates their approach from previous work would improve this section substantially.

line 147-148 "utilization of pulsed mode laser could yield an increase in absorption probability, favoring contingency of regime transitions in laser remelting passes" - the authors cite Singh et al (reference 45) when making this assertion that pulsed melting leads to higher absorptivity. While that manuscript does compare between CW lasers and ns pulsed lasers for LPBF, the most important variable which influences absorptivity discussed in that manuscript is laser wavelength rather than pulse format. It does not specifically claim that absorptivity is higher for ns pulsed lasers than for CW lasers of the same wavelength, as the authors suggest it does. So

A good deal of the discussion on page 8 does not seem relevant to the central claim of the paper. The authors discuss detailed physics of the solidification rates and Marangoni flow, but they do not link these physics to the acoustic emission data that the manuscript is about. It is not clear to me why this discussion is included, in particular since the authors do not appear to be introducing any new physics that have not been previously discussed elsewhere in the literature. It would strengthen the paper if this discussion were truncated to only focus on features relevant to the central claim of the manuscript and/or observations of new physical phenomena that have not been reported previously.

The description of ground truth assignments for stable keyhole vs unstable keyhole sounds like a subjective process based on the description given in the text. The authors should address this possibility and comment on how it affects their results. For example, it is unclear if the approach used here would assign a single frame to the unstable keyhole regime, or if the assignment approach would require multiple consecutive frames to observe the instabilities that characterize this regime.

line 584: "the acoustic data acquisition was manually triggered" - manual triggering of the acoustic data recording implies a lack of explicit timing synchronization between the acoustic signal and the ground truth radiography images. This introduces the potential for significant timing error introduced by a time mismatch between the two datasets. The authors should comment on the uncertainties introduced by this unconventional approach to synchronization, especially when working with data that uses MHz sampling rates.

Reviewer #3 (Remarks to the Author):

In the present work, the authors examine the use of in-situ synchrotron X-ray imaging synchronized with acoustic emission to evaluate and predict the transition between conduction and keyhole defects in LPBF. In my opinion, this research is highly innovative and in general the manuscript is very well-written. Perhaps, the employed technology to predict keyhole defects is too complex to upscale to conventional LPBF machines, but in any case, it deserves to be explored and published in Nature Communications.

Just some minor comments could be addressed before being accepted:

1. Page 4, line 134. It must be explained how the keyhole regime is determined.
2. Figures 1-3. Micrographs at higher magnification or with better resolution must be provided in order to show clearly the features described in the manuscript (page 4), melt pool sizes, unstable keyhole evolution, pore formation etc.
3. Page 8, line 245. How is determined from the X-ray images if the keyhole is stable or unstable?

Reviewer #1

1) *Increasing the understanding of the physics underpinning the LPBF process is critical to its greater adoption by the different industry sectors so I read your paper with much interest. I would like to commend the team on the design of experiments, their detail and depth and the comprehensive and detailed analysis of the results. I have a few comments that you may want to consider before the paper is published.*

We express our gratitude to the reviewer for dedicating his time and effort to reviewing this manuscript. We appreciate the received valuable inputs and will carefully consider them in the revision of the manuscript.

2) *I believe the title needs to be more reflective of the reported work. Currently it is focused on the X-ray imaging but I would focus on AE.*

We agree with the reviewer on his comment on the title of the manuscript. We have carefully chosen an alternative title with more focus on the acoustic aspect of the work.

The revised title is as follows:

"Harmonizing Sound and Light: X-Ray Imaging Unveils Acoustic Signatures of Stochastic Inter-Regime Instabilities during Laser Melting"

3. *Too much emphasis is placed on LPBF when in fact the work is all about laser melting and AE. I note that you have commented on the link between the two but what was not clear to me is whether you collected any AE signals during the manufacture of the SS wall or all the AE signals reported relate to the laser melting of the wall. If this is the case than the paper should just focus on the laser melting experiments and the connection to LPBF should not be that prominent.*

The reason for emphasizing LPBF in our study is to address a common issue that frequently occurs in the LPBF process, particularly in geometries with intricate features. We investigate inter-regime transitions, which predominantly result from varying boundary conditions such as the bulk/surface ratio and dimensional inaccuracies during LPBF. By manufacturing a wall through the LPBF process, we aimed to simulate similar conditions and observe the resulting instabilities that occur during the process.

Additionally, it is important to note that during the LPBF process, the laser vector, apart from the first scan line in each layer, is always partially on the previously solidified track with characteristics of a solid piece fabricated through LPBF. This consideration further reinforces the relevance of our study in understanding the behavior and challenges associated with LPBF.

Although we did not collect acoustic emission (AE) signals during the manufacture of the stainless steel wall, our focused approach enables us to study the specific phenomena that occur during laser melting and their connection to the acoustic emission signals. The AE noise measured during the process is closely related to the behavior of the melt pool and its underlying physics. We find significant similarities between these physics in LPBF and laser remelting, which further strengthens the applicability of our findings.

The connection between LPBF and laser melting is highlighted in our study to demonstrate the relevance and applicability of the findings to the broader LPBF context. By investigating the instabilities and their association with the acoustic emission signals, we aim to contribute to the understanding of the challenges faced in LPBF and provide insights that can potentially improve the process.

We appreciate the reviewer's feedback and have taken it into consideration to carefully use the term LPBF whenever it is relevant and/or replace it with "laser remelting" otherwise.

4. It is not clear to me why is conduction mode observed when the chosen laser parameters, as mentioned in the paper, were meant to only generate the keyhole mode. I think further clarification on this would be helpful.

We appreciate the comment and the opportunity to provide clarification. As stated in the manuscript, the walls were indeed manufactured using laser parameters optimized for the keyhole mode in the LPBF process. However, during the laser remelting passes, where operando X-ray imaging and acoustic emission acquisition were performed, we deliberately employed laser parameters resulting in near-transition conduction conditions. This deliberate choice aimed to allow for the possibility of inter-regime transitions between conduction and keyhole regimes, as even slight variations in the wall's geometry alongside the laser scan vector and the resulting changes in the boundary conditions and heat dissipation could trigger such transitions.

We hope that the provided explanation suffices in clarifying the confusion.

5. When repeating the laser melting scans for individual parameters was the top surface of the wall machined between the scans. The roughness of this surface can significantly affect the absorptivity.

We appreciate the reviewer's comment regarding the potential influence of surface roughness on absorptivity during the repeated laser melting scans. In order to provide representative conditions of the LPBF process, the top surface of the walls was intentionally left unpolished after the laser scan experiments. This decision allows us to capture the natural variations in surface roughness that occur between layers in the LPBF process, which becomes particularly relevant during laser re-melting for various applications within LPBF.

Furthermore, during the laser scanning of the powder bed, only the first scan line processes a region covered by powder, while subsequent scan vectors partially overlap with the previously solidified material based on the hatch distance. By maintaining the surface in this state, we aim to simulate realistic conditions and account for the slight variations in absorptivity and material bulk volume along the laser scanning line.

Furthermore, as illustrated in the image below (showing a section of one of the walls between two laser scans), the variations in height are primarily characterized by high-wavelength waviness, which implies minor fluctuations in absorptivity. In this scenario, the changes in absorptivity are predominantly influenced by the incident angle and do not involve the phenomenon of multiple reflections that may occur on highly rough surfaces with high-frequency waves.

The modified text goes as follows (Main text, section 2.1):

"A non-uniform heat dissipation alongside the wall is induced by keyhole porosities and geometrical unwanted features. Furthermore, the height variations along the wall introduce changes in the laser focus and the projected intensity profile on the top surface of the wall. These height variations, along with the residual waviness left by previous laser remelting, also contribute to variations in the laser incident angle and absorptivity along the track."

6. I suggest changing the laser power to average laser power throughout the text.

We appreciate the suggestion to use the term 'average laser power' instead of 'laser power' throughout the text. While we acknowledge the importance of precise terminology, in our study, we specifically refer to the nominal laser power rather than the average laser power. Therefore, we have made the necessary revisions to consistently use the term 'nominal laser power' in the manuscript, as we did not record the average power of the laser through the laser interface. We value your attention to detail, and your input in ensuring clarity and accuracy in our work is highly appreciated.

7. Can the acoustic signal graphs be made larger. I found them difficult to read in their current version.

We value the feedback provided and acknowledge the importance of readability in the figures. In response, the graphs have been replaced with larger, higher quality images. Additionally, we recommend referring to the figures in the Word version of the document as the auto-generated PDF may not accurately display the high-quality images.

8. Fig 3f has a laser wavelength of 1064nm but the rest of the paper refers to 1070 nm.

We have taken note of the discrepancy and have made the necessary correction. The laser wavelength is now consistently referred to as 1070 nm throughout the paper.

9. Fig 3f also shows that there appears to be lack of close overlap between the duration of the AE signal and the total absorption.

We appreciate the comment and the opportunity to provide clarification. The estimation of total absorption was conducted based on the keyhole cavity's geometry. As the melting regime predominantly exhibited conduction, measurements of total absorption were feasible only during regime instabilities. In the absence of the keyhole, the absorptivity value was assumed to be equal to the Fresnel absorption, which, in this case, is 40%. This assumption accounts for the minimum value on the axis of total absorption in Figure 3f. We hope this explanation helps to address your concern.

10. Table 1. There appears to be misalignment between column headings and columns.

We appreciate the feedback regarding the misalignment issue in Table 1. We have rectified the problem by aligning the column headings with their respective columns.

Reviewer #2

1. This manuscript presents work that is interesting, but is not well suited to the broad audience of Nature Communications and should instead be published in a more specialized journal focused on additive manufacturing. The authors describe experiments that apply operando synchrotron X-ray imaging and acoustic emission monitoring simultaneously to the laser powder bed fusion (LPBF) process. Based on the acoustic monitoring, they train a data-driven model to categorize the LPBF process into 3 different processing regimes: conduction mode, stable keyhole mode, and unstable keyhole mode. They claim that this is the first report of "operando synchrotron X-ray imaging synchronized with acoustic emission recording". If this claim were true, then this work might merit publication in Nature Communications. Unfortunately, it is not. Pandiyan et al (Additive Manufacturing, 2022, 58, 103007) published a previous report on a very similar topic, in which they use operando synchrotron X-ray imaging collected simultaneously with acoustic sensing and other sensing modalities to categorize the laser powder bed fusion process into three process regimes: lack of fusion, conduction mode, and keyhole mode. Due to this extremely similar prior publication, this work is better suited for a more specialized journal.

*We appreciate the reviewer's concern on the novelty of the work, and are happy to confirm that in the present manuscript we do analyze **for the first time** "operando synchrotron X-ray imaging synchronized with acoustic emission recording".*

*The previous publication by Pandiyan et al. is actually our own publication. We therefore know its content very well. The misunderstanding may come from the title, "Deep learning-based monitoring of laser powder bed fusion process on variable time-scales using heterogeneous sensing and operando X-ray radiography guidance", which indeed mentions operando **X-Ray radiography guidance**. The guidance merely consisted in checking that the average melting regimes correlated with the defined process parameters. This check could have been performed from metallographic analysis, i.e. we did not exploit the synchronization of the operando experiments with the acoustic signals. Transitions from one melting regime to another were not observed and studied either (the samples geometry and chosen conditions were in fact not appropriate). The three investigated process regimes corresponded to very different laser parameters, and only the average melting behavior was identified. This explains why the operando experiments were not exploited; they were in fact not needed. The focus was instead placed on developing a hybrid Deep Learning (DL) model, combining Convolutional Neural Networks (CNNs) with Long-Short Term Memory (LSTM), operating over variable time-scales, due to the multiple types of sensors (an acoustic sensor and three photodiode detectors). Furthermore, the acoustic sensor was a structure-borne sensor, very different from the airborne membrane-free optical microphone used in this manuscript.*

*In contrast, in the present work, it is crucial to record and analyze synchronized X-Ray and acoustic information, such as to follow the details of the transitions between melting regimes, at a very fine time scale ($\approx 100 \mu\text{s}$), and occurring under constant laser parameters. **We report it for the first time.** It is only under such conditions that we can isolate the acoustic features specific to each melting regime. This unique (and new) approach clearly demonstrates the relevance of acoustic monitoring for laser based additive manufacturing, whereas in the previous work by Pandiyan et al, and in most of the literature, the classification done by the model to identify the melting regime was based not only on the changes in laser-material interactions, but also on the changes in processing conditions.*

2. Lines 72-87: this section is poorly written and it is not clear what point the authors intend to make. The authors list a variety of hypothetical mechanisms that might lead to acoustic emission during LPBF. But it is not clear if the authors are actually asserting that any of these mechanisms are relevant to their study, and this section would be improved if it were not written in a hypothetical tense. Furthermore, some of the literature the authors cite in this passage of text do not match the claims made. For example in line 72 the authors cite Martin et al (ref 30) to claim "noise is produced during the LPBF process, which can be employed as an in situ feedback signal". This reference does not report acoustic emission measurements or report any attempts at feedback during LPBF.

The passage 'There is considerable acoustic noise generated during the laser-induced melting process, insofar as being used as an in situ process monitoring signal' can be found in the introduction section of ref (30)¹. The authors of the ref (30) cite two publications to support their claims. These publications are as follows:

² *Hongping Gu & W W Duley. A statistical approach to acoustic monitoring of laser welding. J. Phys. D. Appl. Phys. 29, 556 (1996).*

Here the authors report acoustic emission signals from the laser welding process where they demonstrate that the acoustic data is capable of indicating the weld quality, hence carrying information from the laser-material interaction zone.

³ *Everton, S. K., Hirsch, M., Stavroulakis, P. I., Leach, R. K. & Clare, A. T. Review of in-situ process monitoring and in-situ metrology for metal additive manufacturing. Materials and Design at <https://doi.org/10.1016/j.matdes.2016.01.099> (2016).*

Which is principally a review paper where, authors incorporate shortly on the incorporation of acoustic emission in LPBF and Direct Energy Deposition (DED) processes.

We appreciate the reviewer's comment, and we have updated our citation accordingly to ensure the accuracy and relevance of the references in the text. Furthermore, we have

included two additional citations that present a classification of the melting regime based on acoustic data using machine learning and deep learning approaches:

⁴ *Drissi-Daoudi, R. et al. 'Acoustic emission for the prediction of processing regimes in Laser Powder Bed Fusion, and the generation of processing maps.' Addit. Manuf. 67, 103484 (2023).*

⁵ *Drissi-Daoudi, R. et al. 'Differentiation of materials and laser powder bed fusion processing regimes from airborne acoustic emission combined with machine learning.' Virtual Phys. Prototyp. 17, 181–204 (2022).*

As per the reviewer's suggestion, we have made revisions to the text to reduce the use of hypothetical language and provide a more factual and precise description of the phenomena.

The modified text in the manuscript is as follows (Main text, section 1): (the citations in the text below correspond to the citations in the main text)

“During the LPBF process, a significant amount of noise is generated, which can be utilized as an in-situ feedback signal ^{29–31}. In the context of laser welding, acoustic waves can originate from three primary sources: (1) acoustic waves induced within the substrate through phase transformation phenomena, including solid-state, solid/liquid, and liquid/gas transformations ^{32,33} (2) thermal stress-induced acoustic waves in the laser-material interaction zone; and (3) acoustic waves resulting from variations in gas momentum due to heating, boiling processes, and subsequent changes in gas pressure ^{33,34}. The latter can occur due to various phenomena in the laser-material interaction zone, such as metal vaporization, formation and expansion of a microjet at the bottom of the keyhole, pressure variation caused by surface tension gradient around the keyhole opening due to thermal deformation ³⁴, collision of liquid with the keyhole rear wall due to a drop in vapor pressure in the lower part of the cavity, collapse of liquid against the melt pool boundaries, penetration of the microjet into the melt and its impact on the surface of formed keyhole porosities, pore collapse and subsequent rebound, and periodic over-heating induced explosion at the bottom of the keyhole after its collapse ^{8,35–38}. Collectively, these events can contribute to the periodic oscillation of the keyhole within a reported frequency range of 20 – 50 kHz ³⁹.”

3. In lines 74-78, the authors describe 3 potential mechanisms for acoustic emission during laser welding and cite references 32-34. Reference 33 describes shock dynamics and liquid explosions in water droplets caused by X-ray pulses, a substantially different physical process than laser welding. The authors should modify this section to ensure they are citing appropriate references.

The reference (33)⁶ describes the generation of shock dynamics and liquid explosions in water droplets caused by X-ray pulses, which indeed represents a substantially different physical process than laser welding. We apologize for the confusion caused by this reference in the

context of laser welding. To clarify, our intention was to discuss the general concept of acoustic waves generated by variations in gas momentum resulting from heating and boiling processes followed by alterations in gas pressure, which is relevant to laser welding.

Regarding the physical phenomena observed in laser welding, recent advancements in high-speed X-ray imaging techniques have enabled the study of underlying processes. In a notable article by Zhao et al. (*Science* 370, 1080–1086, 2020)⁷, MHz X-ray imaging of the laser-material interaction zone in keyhole mode was conducted, revealing different stages of keyhole generation and collapse in laser welding, such as the formation of metallic jets, fast evaporation-induced shock waves, and cavitation.

Inside the keyhole cavity, the morphology of the keyhole front wall undergoes constant changes with small dynamic bulges moving up and down. These bulges are briefly exposed to laser illumination, leading to rapid and localized heating with limited heat dissipation. As a result, over-heated liquid can reach temperatures above its boiling point, subsequently evaporating with an explosion inside the keyhole cavity, generating acoustic waves⁷. While the reference cited (33)⁶ primarily illustrates the explosion and acoustic wave generation in pure water, it serves to highlight the physical phenomenon of rapid gas momentum variation resulting from heating and boiling processes, which is applicable to liquid metals as well. Therefore, studying and understanding individual micro-events within the LPBF process can provide valuable insights.

To further support the discussion on cavitation events within the melt pool generating their own acoustic waves, we are adding additional citation to the relevant literature. The citation goes as follows:

⁸ Krivokorytov, M. S. et al. Cavitation and spallation in liquid metal droplets produced by subpicosecond pulsed laser radiation. *Phys. Rev. E* 95, 31101 (2017).

The above citation contributes to the understanding of the complex nature of cavitation in liquid metals and their corresponding shock wave.

The modified text in the manuscript is as follows (Main text, section 1): (the citations in the text below correspond to the citations in the main text) – same paragraph as reviewers previous question.

“During the LPBF process, a significant amount of noise is generated, which can be utilized as an in-situ feedback signal^{29–31}. In the context of laser welding, acoustic waves can originate from three primary sources: (1) acoustic waves induced within the substrate through phase transformation phenomena, including solid-state, solid/liquid, and liquid/gas transformations^{32,33} (2) thermal stress-induced acoustic waves in the laser-material interaction zone; and (3) acoustic waves resulting from variations in gas momentum due to heating, boiling processes, and subsequent changes in gas pressure^{33,34}. The latter can occur due to various phenomena in the laser-material interaction zone, such as metal vaporization, formation and expansion of a microjet at the bottom of the keyhole, pressure

variation caused by surface tension gradient around the keyhole opening due to thermal deformation³⁴, collision of liquid with the keyhole rear wall due to a drop in vapor pressure in the lower part of the cavity, collapse of liquid against the melt pool boundaries, penetration of the microjet into the melt and its impact on the surface of formed keyhole porosities, pore collapse and subsequent rebound, and periodic over-heating induced explosion at the bottom of the keyhole after its collapse^{8,35-38}. Collectively, these events can contribute to the periodic oscillation of the keyhole within a reported frequency range of 20 – 50 kHz³⁹.”

4. Line 105: "for the first time" - as mentioned earlier, this statement is incorrect as Pandiyan et al (reference 27) reported very similar work in 2022.

As detailed in the answer to question 1, we confirm that the two papers are very different, and that the elements claimed to be novel in the present paper were not part of our previous work published in 2022.

However we agree with the reviewer that lines 105-107 are not fully accurate. We have therefore added the word "synchronized" (in red).

The modified text goes as follows (Main text, section 1):

*"A combination of operando high-speed X-ray imaging and implementation of **synchronized** in-situ acoustic measurements could, for the first time, reveal the critical relationship between the physical events and the corresponding acoustic signal frequency contents."*

*The synchronized X-Ray imaging and in-situ acoustic measurements were indeed used and analyzed **for the first time** in this paper. In doing so, we revealed, for the first time, the critical relationship between the physical events and the corresponding acoustic signal frequency contents. In contrast, and as discussed in the answer to question 1, previous work from literature, including our own (reference 27), only classified the average melting regime induced by different processing conditions. And it was not possible to isolate the acoustic features exclusively related to the laser-material interaction of a given melting regime: process conditions always interfered. In reference 27, as noted earlier, synchronized operando X-Ray imaging was not exploited, and in fact not even needed, considering the scope of the paper.*

5. line 111: "a data driven filtering pipeline" - it is unclear how the "data driven filtering" approach the authors mention here and describe in the supplementary information is substantially different than "machine learning" or "artificial intelligence" that the authors claim to be fundamentally flawed in the previous paragraph. While the specific architecture of their data driven model is different than the previous work the authors disparage above, they still employ a training-based model that is susceptible to the same challenges many

machine learning approaches face when differentiating between highly correlated signatures that arise from complicated physical system such as LPBF. A more nuanced discussion of what differentiates their approach from previous work would improve this section substantially.

We appreciate the reviewer's point about clarifying the distinction between our proposed data-driven filtering pipeline and the typical machine learning or artificial intelligence approaches we critique earlier in the text. It is correct that both methods seek to extract meaningful information from complex datasets, but their objectives and operations diverge significantly.

Traditional machine learning methods, as have been applied in the previous studies we mentioned, typically focus on classification tasks. They are trained to identify distinct categories in the dataset — here, different process regimes or defect formations. The output of such models is a label or category assignment for each data point or group of points. However, as the reviewer points out, this approach can struggle with highly correlated signatures and may learn to detect changes in process parameters rather than the desired phenomena.

*Our proposed filtering pipeline, on the other hand, is designed to perform segmentation rather than classification. We are not simply assigning categories to data points; we are using the filters to transform the raw acoustic emission signal into three distinct signals, each representing one of the three melting regimes. The softmax function then converts these into a probability distribution across the regimes for each data point. This approach provides a **continuous**, probabilistic assessment of the melting regime throughout the process, not just a **discrete** classification. It is therefore ideal to analyze sudden changes in melting regimes occurring over very short time scales ($\approx 100 \mu\text{s}$). The interpretation of the softmax outputs then directly relates to the physical phenomena we are investigating, offering a more interpretable understanding of the process dynamics, almost impossible to achieve with traditional machine learning methods.*

While we acknowledge that our approach still involves model training and may face similar challenges as machine learning methods, we believe the fundamental shift from classification to segmentation, and the incorporation of physical knowledge in the model design, constitutes a significant advancement over the existing methods. We hope this clarifies our position and have added this explanation to the revised manuscript, to help readers better understand our methodology.

The modified text in the manuscripts is as follows (Main text, section 1): (the citations in the text below correspond to the citations in the main text)

"Our research aims to delve deeper into the LPBF process by focusing on the identification of individual events of defect formation and regime changes, particularly under constant

process parameters. This novel approach seeks to uncover specific acoustic signatures associated with these events, enhancing the robustness of AE-based monitoring — an aspect that has received limited attention in the existing literature. In our innovative methodology, we have introduced, for the first time, the simultaneous synchronized use of operando high-speed X-ray imaging and in-situ acoustic measurements. Specifically, we have observed inter-regime instabilities in LPBF fabricated structures under constant process parameters and conditions using operando high-speed transmission X-ray imaging. Concurrently, integrating an optical microphone into the LPBF building chamber allows us to capture in-situ airborne acoustic emission signals during the process. This methodology enables us to reveal the critical relationship between physical events and their corresponding acoustic signal frequency contents.

Distinct from traditional machine learning classification models ²⁷, our work adopts a segmentation-based approach that enables continuous, probabilistic assessment of the melting regime throughout the process, not just a discrete classification. Our novel contribution lies in the introduction of a data-driven filtering pipeline that effectively segments the acoustic emission signals into corresponding regimes, utilizing the X-ray imaging data as ground truth. This technique, although data-driven, differs fundamentally from typical machine learning or artificial intelligence algorithms. While the latter often relies on extensive training with a wide range of data, and its performance can be hindered by changes in the data distribution or highly correlated features, our method is designed to be robust under constant process parameters, with an emphasis on identifying the correlation between individual physical events and their acoustic signatures. The data-driven filtering pipeline is tailored to cope with the complex and high-dimensional LPBF process by leveraging the X-ray imaging data, which provides an explicit physical interpretation to guide the segmentation. Therefore, we believe this approach is more immune to the challenges many machine learning methods face when differentiating between highly correlated signatures from complex physical systems such as LPBF. Moreover, an essential advantage of our approach is the simplicity of the filtering pipeline, which enables a physically-backed interpretation of the results. This straightforwardness paves the way for the development of robust acoustic emission-based in-situ monitoring systems capable of accurately capturing stochastic regime instabilities and porosity formation.

We believe that our experiments signify a crucial advancement in the understanding of laser melting processes. These insights will contribute to developing practical monitoring techniques, promoting the broader adoption and implementation of metal additive manufacturing technologies in demanding industrial sectors. We hope that by enhancing the reliability and predictability of the laser melting processes through real-time monitoring, we will facilitate its widespread implementation in strategic industries, including aerospace, automobile, and medical. The knowledge gained from this study has profound implications for the future of advanced manufacturing and opens up avenues for further research and development in this field.”

6. line 147-148 "utilization of pulsed mode laser could yield an increase in absorption probability, favoring contingency of regime transitions in laser remelting passes" - the authors cite Singh et al (reference 45) when making this assertion that pulsed melting leads to higher absorptivity. While that manuscript does compare between CW lasers and ns pulsed lasers for LPBF, the most important variable which influences absorptivity discussed in that manuscript is laser wavelength rather than pulse format. It does not specifically claim that absorptivity is higher for ns pulsed lasers than for CW lasers of the same wavelength, as the authors suggest it does.

We appreciate the reviewer's comment and clarification regarding the assertion made in our manuscript. Upon re-evaluation, we acknowledge that our statement could have been more precise.

When comparing pulsed lasers to continuous lasers, it is important to note that pulsed lasers typically have lower average power but higher peak power in each pulse. This characteristic results in higher power density and a greater probability of creating a depression during laser welding. The higher peak power allows for rapid heat transfer to the material, facilitating the formation of a keyhole.¹⁰

¹⁰ *Coroado, J. et al. Comparison of continuous and pulsed wave lasers in keyhole welding of stainless-steel to aluminium. Int. J. Adv. Manuf. Technol. 119, 367–387 (2022).*

Although the statement above refers to studies involving nano-pulsed lasers, the underlying physics can still be applicable to pulsed lasers with repetition rates as low as 250 kHz. The higher peak powers generated by pulsed lasers can have significant effects on the laser-material interaction zone, where boundary conditions are designed to promote non-uniform heat dissipation along the laser track.

We have replaced the previous citation with the one presented above in the manuscript¹⁰ (Citation 47 in the main text).

7. A good deal of the discussion on page 8 does not seem relevant to the central claim of the paper. The authors discuss detailed physics of the solidification rates and Marangoni flow, but they do not link these physics to the acoustic emission data that the manuscript is about. It is not clear to me why this discussion is included, in particular since the authors do not appear to be introducing any new physics that have not been previously discussed elsewhere in the literature. It would strengthen the paper if this discussion were truncated to only focus on features relevant to the central claim of the manuscript and/or observations of new physical phenomena that have not been reported previously.

We appreciate the reviewer's comment on the relevance of the discussion in page 8. We would like to provide clarification on the inclusion of this discussion in relation to the central claim of the manuscript.

The manuscript presents two key novelties: Firstly, it presents the first-ever operando observation of inter-regime instabilities at constant process parameters during the LPBF process, which are responsible for stochastic defects even under optimized process parameters, hindering the attainment of a defect-free state. The second novelty lies in the synchronous recording of x-ray imaging and acoustic emission to capture the acoustic signature of these inter-regime instabilities utilizing high time resolution ground truth data.

The discussion on page 8 provides valuable information on the short- or moderate-lived phenomena occurring during the process and delves deeper into the collateral effects of these instabilities. Specifically, it explores the changes in melt pool and depression zone dimensions, providing further insight into the extent of these transitions' impact. During these transitions, sudden changes in absorptivity and corresponding heat input induce variations in the magnitude and direction of the velocity tensor of the solid-liquid boundary. These variations have established effects on the local microstructure. Additionally, we employ a simple model to estimate the number of reflections based solely on the depression geometry and its correlation with the amplitude of the raw acoustic signal.

While some of the physics discussed in this section may have been previously addressed in the literature, the intention here is to provide a comprehensive understanding of the complex dynamics and implications of the inter-regime instabilities within the context of the acoustic emission data. By linking these phenomena to the observed acoustic signatures, we enhance our understanding of the interplay between the physics of the process and the resulting acoustic emissions. With an X-ray imaging acquisition rate of 10 kHz, we successfully localized significant events such as regime transitions and keyhole collapse in both space and time, allowing us to correlate these events with the acoustic signatures captured by the microphone. However, to establish links between less significant and short-lived events, a higher X-ray imaging acquisition rate closer to that of the acoustic signal is required. This higher time resolution would enable us to capture and analyze these rapid events more effectively, providing a more comprehensive understanding of the laser melting process and its associated acoustic emissions.

8. The description of ground truth assignments for stable keyhole vs unstable keyhole sounds like a subjective process based on the description given in the text. The authors should address this possibility and comment on how it affects their results. For example, it is unclear if the approach used here would assign a single frame to the unstable keyhole regime, or if the assignment approach would require multiple consecutive frames to observe the instabilities that characterize this regime.

We appreciate the reviewer's question regarding how the stability or instability of the keyhole is determined from the X-ray images. In our study, the transition from a stable to an unstable keyhole was labeled when the keyhole started to exhibit instability through periodic collapse and the formation of porosities.

It is worth noting that the reported keyhole fluctuation occurs at a frequency of approximately 20-50 kHz. To capture these collapse events, X-ray imaging at frame rates well above 100 kHz is required. As our current X-ray acquisition frequency does not reach such high frame rates, we adopted a systematic approach based on the first appearance of a porosity resulting from a keyhole collapse. These pores tend to remain visible and reliable for a longer duration, allowing us to observe and analyze them in the current X-ray acquisition setup. Therefore, the first image in which a pore appears is indexed as the start of unstable keyhole and consequently the moment the generation of the pores ceases to occur in the presence of depression is indexed as stable keyhole.

The time resolution of one X-ray frame (100 μ s) introduces a certain level of uncertainty in precisely determining the boundaries between the stable and unstable keyhole regimes. This uncertainty is approximately equal to the duration of one X-ray frame.

The added paragraph goes as follows (Main text, section 2.1):

"The stable keyhole and unstable keyhole were distinguished based on observable characteristics in the X-ray images. A stable keyhole is characterized by a continuous, well-defined shape and absence of significant fluctuations or instabilities. On the other hand, an unstable keyhole exhibits periodic collapse and the formation of porosities. These instabilities are visible as changes in the keyhole shape, such as irregular boundaries, fluctuating depths, and the presence of porosity formations. To determine the transition from stable to unstable keyhole, we focused on the first appearance of a porosity resulting from a keyhole collapse. The appearance of porosities indicates the onset of instability in the keyhole, marking the transition to the unstable regime. Conversely, the stable keyhole regime is identified when the generation of porosities ceases and the keyhole remains relatively stable without significant fluctuations. It is worth noting that the time resolution of one X-ray frame (100 μ s) introduces a potential uncertainty when precisely indexing the stable and unstable keyhole regimes. This uncertainty is associated with the duration of one X-ray frame and may affect the exact determination of the transition between these regimes."

9. line 584: "the acoustic data acquisition was manually triggered" - manual triggering of the acoustic data recording implies a lack of explicit timing synchronization between the acoustic signal and the ground truth radiography images. This introduces the potential for significant timing error introduced by a time mismatch between the two datasets. The authors should comment on the uncertainties introduced by this unconventional approach to synchronization, especially when working with data that uses MHz sampling rates.

To address these concerns, we would like to highlight some important observations. In our experiments, the XARION optical microphone was triggered manually due to the unavailability of a feasible method for triggering the microphone through a TTL signal from the laser. However, the acquired acoustic signals exhibited a remarkably high signal-to-noise ratio, as evident in Figure a (below). The clear and distinct intersections observed in the signal indicate the beginning and end of the process.

Upon closer examination of the data in Figures b and c, it is apparent that the amplitude changes at the beginning and end of the process remain discernible. Although a zone of uncertainty is observed at the end of the process, with a duration of up to 100 μ s, it is important to note that this uncertainty is equal to the time resolution of the x-ray images. Thus, the effect of the manual triggering on the timing synchronization is minimal during these experiments.

Furthermore, it is worth mentioning that even if the acoustic data triggering were done with the laser TTL signal, there would still be a delay caused by the distance between the microphone and the process zone, considering the sound propagation in the gaseous medium. In our setup, the microphone was placed at a distance of 0.1 m from the process zone in an Argon atmosphere. Considering the sound velocity in Argon to be approximately 323 m/s, it would take around 310 μ s for the sound to reach the microphone from the process zone. This delay is more than three times the current uncertainty observed in the manual synchronization.

We hope that these clarifications and analysis provide a clearer understanding of the accuracy and limitations associated with the manual synchronization approach in our study.

The added information goes as follows (Main text, section 4.3.2):

“Upon closer examination of the acoustic amplitude changes at the beginning and end of the laser scan, it was observed that there is a region of uncertainty spanning approximately 100 μ s. It is important to note that this uncertainty aligns with the time resolution of the X-ray imaging, which indicates that the precise indexing of the stable and unstable keyhole regimes may have an associated uncertainty of approximately 100 μ s.”

Reviewer #3

1. In the present work, the authors examine the use of in-situ synchrotron X-ray imaging synchronized with acoustic emission to evaluate and predict the transition between conduction and keyhole defects in LPBF. In my opinion, this research is highly innovative and in general the manuscript is very well-written. Perhaps, the employed technology to predict keyhole defects is too complex to upscale to conventional LPBF machines, but in any case, it deserves to be explored and published in Nature Communications.

Just some minor comments could be addressed before being accepted:

We sincerely appreciate the reviewer's thoughtful comments and positive assessment of our work. It is encouraging to receive feedback acknowledging the innovative nature of our research and the overall quality of the manuscript.

2. Page 4, line 134. It must be explained how the keyhole regime is determined.

This determination was confirmed through microstructural analysis and observation of the melt pool geometry. However, it is important to note that the process optimization was specifically focused on the fabrication of the walls using the laser at the focal point, and as such, we did not present a process map in this particular study. Thank you for bringing this to our attention and we have updated the text in the supplementary material to address this lack of clarity.

The added sentence goes as follows (Supplementary method 3):

"The keyhole regime was confirmed through ex-situ microstructural analysis and observation of the melt pool geometry."

3. Figures 1-3. Micrographs at higher magnification or with better resolution must be provided in order to show clearly the features described in the manuscript (page 4), melt pool sizes, unstable keyhole evolution, pore formation etc.

In response to this feedback, we have updated the graphs in the main text and supplementary material with maximum resolution micrographs that provide better clarity and detail. Additionally, we have included larger figures for the acoustic emission data to enhance their visibility. We encourage the reviewers to refer to the word version of the document rather than the auto-generated PDF provided by the journal to take full advantage of the high-resolution images and figures.

Thank you for your valuable suggestion, and we appreciate your attention to detail in helping us enhance the quality of our work.

4. Page 8, line 245. How is determined from the X-ray images if the keyhole is stable or unstable?

We appreciate the reviewer's question regarding how the stability or instability of the keyhole is determined from the X-ray images. In our study, the transition from a stable to an unstable keyhole was labeled when the keyhole started to exhibit instability through periodic collapse and the formation of porosities.

It is worth noting that the reported keyhole fluctuation occurs at a frequency of approximately 20-50 kHz. To capture these collapse events, X-ray imaging at frame rates well above 100 kHz is required. As our current X-ray acquisition frequency does not reach such high frame rates, we adopted a systematic approach based on the first appearance of a porosity resulting from a keyhole collapse. These pores tend to remain visible and reliable for a longer duration, allowing us to observe and analyze them in the current X-ray acquisition setup. Therefore, the first image that a pore appears is indexed as the start of unstable keyhole and consequently the moment the generation of the pores ceases to exist in the presence of depression is indexed as stable keyhole.

The time resolution of one X-ray frame ($\approx 100 \mu\text{s}$) introduces a certain level of uncertainty in precisely determining the boundaries between the stable and unstable keyhole regimes. This uncertainty is approximately equal to the duration of one X-ray frame.

The added paragraph goes as follows (Main text, section 2.1):

"The stable keyhole and unstable keyhole were distinguished based on observable characteristics in the X-ray images. A stable keyhole is characterized by a continuous, well-defined shape and absence of significant fluctuations or instabilities. On the other hand, an unstable keyhole exhibits periodic collapse and the formation of porosities. These instabilities are visible as changes in the keyhole shape, such as irregular boundaries, fluctuating depths, and the presence of porosity formations. To determine the transition from stable to unstable keyhole, we focused on the first appearance of a porosity resulting from a keyhole collapse. The appearance of porosities indicates the onset of instability in the keyhole, marking the transition to the unstable regime. Conversely, the stable keyhole regime is identified when the generation of porosities ceases and the keyhole remains relatively stable without significant fluctuations. It is worth noting that the time resolution of one X-ray frame ($100 \mu\text{s}$) introduces a potential uncertainty when precisely indexing the stable and unstable keyhole regimes. This uncertainty is associated with the duration of one X-ray frame and may affect the exact determination of the transition between these regimes."

Bibliography

1. Martin, A. A. *et al.* Ultrafast dynamics of laser-metal interactions in additive manufacturing alloys captured by in situ X-ray imaging. *Mater. Today Adv.* **1**, 100002 (2019).
2. Hongping Gu & W W Duley. A statistical approach to acoustic monitoring of laser welding. *J. Phys. D. Appl. Phys.* **29**, 556 (1996).
3. Everton, S. K., Hirsch, M., Stavroulakis, P. I., Leach, R. K. & Clare, A. T. Review of in-situ process monitoring and in-situ metrology for metal additive manufacturing. *Materials and Design* at <https://doi.org/10.1016/j.matdes.2016.01.099> (2016).
4. Drissi-Daoudi, R. *et al.* Acoustic emission for the prediction of processing regimes in Laser Powder Bed Fusion, and the generation of processing maps. *Addit. Manuf.* **67**, 103484 (2023).
5. Drissi-Daoudi, R. *et al.* Differentiation of materials and laser powder bed fusion processing regimes from airborne acoustic emission combined with machine learning. *Virtual Phys. Prototyp.* **17**, 181–204 (2022).
6. Stan, C. A. *et al.* Liquid explosions induced by X-ray laser pulses. *Nat. Phys.* **12**, 966–971 (2016).
7. Zhao, C. *et al.* Critical instability at moving keyhole tip generates porosity in laser melting. *Science (80-.).* **370**, 1080–1086 (2020).
8. Krivokorytov, M. S. *et al.* Cavitation and spallation in liquid metal droplets produced by subpicosecond pulsed laser radiation. *Phys. Rev. E* **95**, 31101 (2017).
9. Pandiyan, V. *et al.* Deep learning-based monitoring of laser powder bed fusion process on variable time-scales using heterogeneous sensing and operando X-ray radiography guidance. *Addit. Manuf.* **58**, 103007 (2022).
10. Coroado, J. *et al.* Comparison of continuous and pulsed wave lasers in keyhole welding of stainless-steel to aluminium. *Int. J. Adv. Manuf. Technol.* **119**, 367–387 (2022).

REVIEWERS' COMMENTS

Reviewer #1 (Remarks to the Author):

Thank you for addressing my comments in detail which have improved the paper. I think the paper is novel and the results contribute to not only increased understanding of the physics behind the LPBF process but also potential process qualification approach. I believe the paper can be published as is.

Milan Brandt

Reviewer #3 (Remarks to the Author):

The revised version, together with the information provided in the response to the referees met all concerns of this reviewer. The paper is suitable for publication.